# VIPO: Value Function Inconsistency Penalized Offline Reinforcement Learning

**Xuyang Chen** [1]  **Keyu Yan** [1]  **Guojian Wang** [1]  **Lin Zhao** [1]

## Abstract

Offline reinforcement learning (RL) learns effective policies from pre-collected datasets, offering a practical solution for applications where online interactions are risky or costly. Model-based approaches are particularly advantageous for offline RL, owing to their data efficiency and generalizability. However, due to inherent model errors, model-based methods often artificially introduce conservatism guided by heuristic uncertainty estimation, which can be unreliable. In this paper, we introduce VIPO, a novel model-based offline RL algorithm that incorporates self-supervised feedback from value estimation to enhance model training. Specifically, the model is learned by additionally minimizing the inconsistency between the value learned directly from the offline data and the value estimated from the model. We perform comprehensive evaluations from multiple perspectives to show that VIPO can learn a highly accurate model efficiently and consistently outperform existing methods. In particular, it achieves state-of-the-art performance on almost all tasks in both D4RL and NeoRL benchmarks. Overall, VIPO offers a *general framework* that can be readily integrated into existing model-based offline RL algorithms to systematically enhance model accuracy. Our code is available at https://github.com/NUS-CORE/vipo.

## 1. Introduction

Offline reinforcement learning (RL) (Lange et al., 2012; Levine et al., 2020) aims to learn effective policies exclusively from a pre-collected behavior dataset, eliminating the need for online interaction with the environment. This approach offers a compelling solution for tasks where online interactions entail significant risks or exorbitant costs. Due

to its potential to transform static datasets into powerful decision-making systems, offline RL has gained popularity in recent years (Kalashnikov et al., 2018; Prudencio et al., 2023). While off-policy RL algorithms can in principle be directly applied to the offline datasets, recent studies show that they can perform poorly when applied offline (Fujimoto et al., 2019; Kumar et al., 2019). This is primarily attributed to the distribution shift between the learned and the behavior policies, as learning policies beyond the behavior policy requires querying the actions not observed in the dataset. Similarly, value function evaluation on out-of-distribution (OOD) actions is usually inaccurate. In turn, maximizing an inaccurate value function during policy improvement often overestimates the value of OOD actions, which is a core challenge in offline RL.

To mitigate overestimation, a common paradigm in offline RL is to incorporate conservatism into algorithm design. Model-free offline RL algorithms (Fujimoto et al., 2018; Wu et al., 2019; Kumar et al., 2020; Wang et al., 2022) achieve this by learning a pessimistic value function (value-based approach) to discourage choosing OOD actions or constraining the learned policy (policy-based approach) to avoid visiting such actions. However, these algorithms often suffer from excessive conservative, as many of them solely learn from the dataset (Wang et al., 2018; Chen et al., 2020; Kostrikov et al., 2021). Meanwhile, it is crucial to balance conservatism and generalization since being overly conservative hinders finding a better policy. In contrast, model-based offline RL algorithms (Kidambi et al., 2020; Yu et al., 2020; Sun et al., 2023) ensure conservatism by learning a pessimistic dynamics model, which penalizes the value of OOD actions. Additionally, model-based algorithms have the potential for broader generalization as they can incorporate dynamic models to generate synthetic data that are not present in the dataset (Yu et al., 2020). This advantage makes model-based methods particularly well-suited for offline RL, motivating their further investigation in this paper.

Most prior model-based algorithms rely on uncertainty estimation to incorporate conservatism. They commonly train an ensemble of $N$ models through maximum likelihood on the dataset and quantify uncertainty based on the standard deviation among the predictions of these $N$ models. Specifically, Yu et al. (2020) employs the max-aleatoric uncertainty

[1]Department of Electrical and Computer Engineering, National University of Singapore, Singapore. Correspondence to: Lin Zhao <elezhli@nus.edu.sg>.

*Proceedings of the 43rd International Conference on Machine Learning*, Seoul, South Korea. PMLR 306, 2026. Copyright 2026 by the author(s).

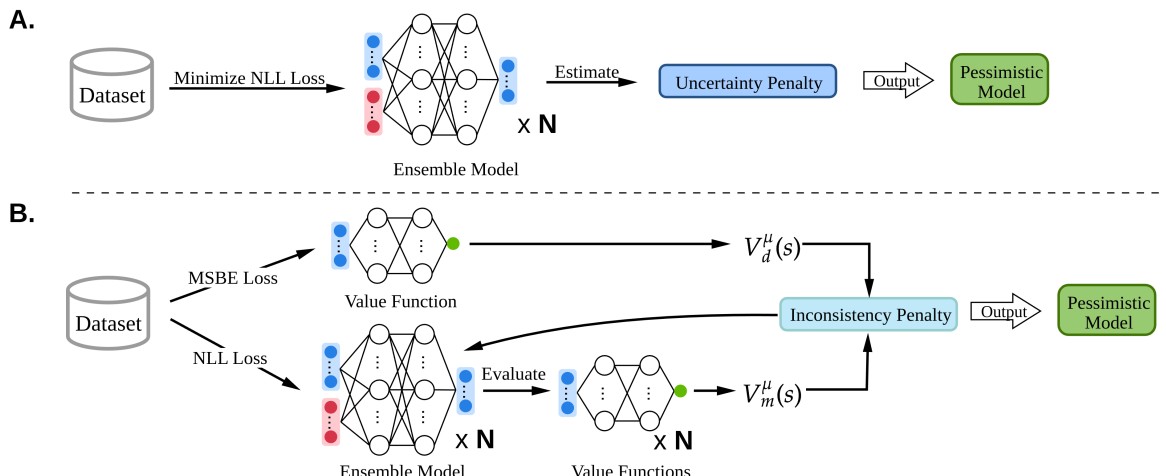

*Figure 1.* Comparison of VIPO with previous model-based approaches for learning a pessimistic dynamics model. **(A)** Previous model-based methods make a single use of data to learn an ensemble of models and then use its uncertainty to apply *ad-hoc*, pessimistic adjustments to the model predictions. **(B)** VIPO leverages the data in two ways: (1) it learns a value function $V_d^\mu(s)$ directly by minimizing the mean square Bellman error (MSBE) loss; (2) it first learns the dynamics model by minimizing the negative log-likelihood (NLL) loss and then estimates the ensemble value functions $V_m^\mu(s)$. VIPO utilizes the discrepancy between these two types of value functions as an additional self-supervised loss to improve the model learning performance.

quantifier, Kidambi et al. (2020) adopts the max-pairwise-diff uncertainty quantifier, and Lu et al. (2021) utilizes the ensemble-standard-deviation uncertainty quantifier. Nevertheless, it was shown that the uncertainties of their learned models using these methods are inaccurate and unreliable when dealing with complex datasets in practice (Ovadia et al., 2019). This aligns with our findings: as shown empirically in Section 5.2, even in Gym tasks the uncertainties estimated by MOPO-learned models fail to reflect the expected increase as the data-drop ratio grows, whereas those from VIPO-learned models track this trend well. This indicates that VIPO captures epistemic uncertainty due to data loss more effectively than MOPO and thus is likely to learn a more accurate model.

In this paper, we introduce **V**alue Function **I**nconsistency **P**enalized **O**ffline Reinforcement Learning (VIPO), a model-based offline RL algorithm that aims to learn a highly accurate model by integrating a value function inconsistency loss into model training. VIPO differs fundamentally from prior methods in two key aspects. First, our approach makes a dual-usage of the data in a self-supervised fashion to improve the model training accuracy. As illustrated in Figure 1, previous model-based RL methods make only a single-purpose use of the data: they learn an ensemble of models and obtain a pessimistic model by applying ad-hoc adjustments to the model outputs based on the uncertainty estimated from the ensemble. In contrast, VIPO leverages the data in two complementary ways. The first use of the data is to directly learn an approximated value function of the behavior policy by minimizing the empirical Mean Square Bellman Error (MSBE). The second use of the data is to learn a model by minimizing the negative log-likelihood

loss (NLL), followed by value evaluation to obtain another approximated value function of the behavior policy. If the learned model is accurate, the two value functions should coincide owing to the *uniqueness* of the value function, and their alignment (or lack thereof) provides a quantifier of model accuracy. Since offline RL must rely entirely on the fixed dataset, effectively exploiting it is critical. Compared with prior approaches that use the data for a single purpose, our method leverages it in two complementary ways.

Second, unlike previous works which do not mitigate the ensemble-model uncertainty in the training stage, we proactively incorporate the value function inconsistency into the model training loss in a self-supervised manner to improve model learning. By establishing the analytical expression for the gradient of the augmented loss in Theorem 4.5, we enable the model to be trained via gradient descent, ensuring a balance between data fidelity and alignment in value functions. To the best of our knowledge, no existing model-based offline RL method has incorporated either uncertainty or inconsistency into the training stage. Our approach is the first to proactively mitigate inconsistency during training, instead of ad-hoc adjustments of the model predictions.

Our empirical results demonstrate from multiple perspectives that incorporating value function inconsistency into model error enhances model accuracy, highlighting the effectiveness of the aforementioned two novel designs. VIPO offers a general framework that can be readily integrated into existing model-based offline RL algorithms to systematically enhance model accuracy. We achieve this without relying on expensive architectures such as diffusion or transformer, rather than by comprehensive exploitation of data.

Moreover, VIPO outperforms prior offline RL algorithms and achieves state-of-the-art (SOTA) performance on most tasks of both D4RL and NeoRL benchmarks.

## 2. Related Work

Offline RL focuses on learning effective policies solely from a pre-collected behavior dataset and has demonstrated significant success in practical applications (Rafailov et al., 2021; Singh et al., 2020; Li et al., 2010; Yu et al., 2018). Similar to online RL, offline RL has been explored using both model-free and model-based algorithms, distinguished by whether or not they involve learning a dynamics model.

**Model-free offline RL.** Existing model-free methods on offline RL can be roughly categorized into the following two taxonomies: pessimistic value-based methods and regularized policy-based methods. Pessimistic value-based approaches achieve conservatism by incorporating penalty terms into the value optimization objective, discouraging the value function from being overly optimistic on out-of-distribution (OOD) actions. Specifically, CQL (Kumar et al., 2020) applies equal penalization to Q-values for all OOD samples, whereas EDAC (An et al., 2021) and PBRL (Bai et al., 2022) adjust the penalization based on the uncertainty level of the Q-value, measured using a neural network ensemble. In comparison, regularized policy-based approaches constrain the learned policy to stay close to the behavior policy, thereby avoiding OOD actions. For instance, BEAR (Kumar et al., 2019) constrains the optimized policy by minimizing the MMD distance to the behavior policy. BCQ (Fujimoto et al., 2019) restricts the action space to those present in the dataset by utilizing a learned Conditional-VAE (CVAE) behavior-cloning model. Alternatively, TD3+BC (Fujimoto & Gu, 2021) simply adds a behavioral cloning regularization term to the policy optimization objective and achieves excellent performance across various tasks. IQL (Kostrikov et al., 2021) adopts an advantage-weighted behavior cloning approach, learning Q-value functions directly from the dataset. Meanwhile, DQL (Wang et al., 2022) leverages diffusion policies as an expressive policy class to enhance behavior-cloning.

**Model-based offline RL.** We focus on Dyna-style model-based RL (Janner et al., 2019), which learns a dynamics model from the dataset and uses it to augment the dataset with synthetic samples. However, due to inevitable model errors, conservatism remains crucial to prevent the policy from overgeneralizing to regions where the dynamics model predictions are unreliable. For example, COMBO (Yu et al., 2021) extends CQL to a model-based setting by enforcing small Q-values for OOD samples generated by the dynamics model. RAMBO (Rigter et al., 2022) incorporates conservatism by adversarially training the dynamics model to minimize the value function while maintaining accurate transition predictions. Most model-based methods achieve conservatism through uncertainty quantification, penalizing rewards in regions with high uncertainty. Specifically, MOPO (Yu et al., 2020) uses the max-aleatoric uncertainty quantifier, MOReL (Kidambi et al., 2020) employs the max-pairwise-diff uncertainty quantifier, and MOBILE (Sun et al., 2023) leverages the Model-Bellman inconsistency uncertainty quantifier. Distinct from methods relying on explicit uncertainty quantification, LEQ (Park & Lee, 2024) induces conservatism via lower expectile regression of $\lambda$-returns to enable low-bias model-based value estimation. In this work, we achieve conservatism by incorporating the value function inconsistency loss, enabling the training of a more reliable model.

## 3. Preliminaries

### 3.1. Offline Reinforcement Learning

RL problems are commonly formulated within the framework of a Markov Decision Process (MDP), defined by the tuple $\mathcal{M} = (\mathcal{S}, \mathcal{A}, r, \rho_0, P, \gamma)$. Here, $\mathcal{S}$ denotes the state space, $\mathcal{A}$ represents the action space, $r(s, a) : \mathcal{S} \times \mathcal{A} \to [-r_{\max}, r_{\max}]$ is a bounded reward function, $\rho_0(s)$ specifies the initial state distribution, $P : \mathcal{S} \times \mathcal{A} \to \Delta(\mathcal{S})$ defines the transition kernel, and $\gamma \in [0, 1)$ is the discount factor. Given a policy $\pi(\cdot \mid s)$, its performance can be evaluated by the expected cumulative long-term reward, defined as

$$J(\pi) = \mathbb{E}_{s_0 \sim \rho_0, a_t \sim \pi, s_{t+1} \sim P} \left[ \sum_{t=0}^{\infty} \gamma^t r(s_t, a_t) \right]. \quad (1)$$

The goal of RL is to learn a policy $\pi(\cdot \mid s)$ that maximizes the expected cumulative long-term reward, i.e., $\pi^* = \arg\max_\pi J(\pi)$.

As shown in Eq. (1), the classical RL framework requires online interactions with the environment $P$ during training. In contrast, the offline RL learns only from a fixed offline dataset $D = \{(s_i, a_i, r_i, s_i')\}_{i=1}^N$ collected by the behavior policy $\mu(\cdot \mid s)$, where $s, a, r$, and $s'$ denote the state, action, reward, and next state, respectively. That is, it aims to find the *best possible* policy solely from $\mathcal{D}$ without additional interactions with the environment.

### 3.2. Model-based Offline RL Algorithms

Model-based offline RL aims to derive the optimal policy by utilizing a learned dynamics model. These methods define a pessimistic MDP $\widehat{\mathcal{M}} = \langle \mathcal{S}, \mathcal{A}, \hat{r}, \rho_0, \hat{P}, \gamma \rangle$, which shares the same state and action spaces as the original MDP but employs the learned transition dynamics $\hat{P}(s' \mid s, a)$ and reward function $\hat{r}(s, a)$. $\hat{P}$ and $\hat{r}$ are typically estimated through supervised regression on the dataset $\mathcal{D}$, with additional incorporation of uncertainty penalties to discourage exploration of OOD actions. These pessimistic models then

serve as proxies for the real environment, enabling the simulation of transitions and subsequent use for planning.

The success of model-based RL algorithms hinges on the ability to learn an accurate model. Once the model is learned, various approaches, such as model predictive control (MPC) (Williams et al., 2017), dynamic programming (Munos & Szepesvári, 2008), or model-based policy optimization (Janner et al., 2019), can be employed to recover the optimal policy by solving the pessimistic MDP $\widehat{\mathcal{M}}$.

### 3.3. General Function Approximation.

We study offline RL within the framework of general function approximation. Specifically, we represent the environment dynamics using a conditional model class $\mathcal{P} := \{P_\theta\}_{\theta \in \Theta}$, where each $P_\theta(s', r | s, a)$ models the joint distribution of the next state and reward. We assume this class is sufficiently expressive to capture complex environmental structures, utilizing parameterized families such as neural networks. To facilitate our theoretical analysis, we quantify the complexity of the hypothesis spaces using the $1/N$-bracketing number, denoted by $\mathcal{N}_\mathcal{P}(1/N)$ for the model class and $\mathcal{N}_\mathcal{V}(1/N)$ for the loss class induced by the value function inconsistency (Geer, 2000).

## 4. Our Method

### 4.1. Value Function Learning

We first learn an approximated value function of the behavior policy directly from the dataset. In the MDP $\mathcal{M} = (\mathcal{S}, \mathcal{A}, r, \rho_0, P, \gamma)$, under the behavior policy $\mu(\cdot | s)$, the Bellman backup for obtaining the corresponding value function $V$ is defined as

$$\mathcal{T}^\mu V(s) := \mathbb{E}_{a \sim \mu(\cdot | s), s' \sim P(\cdot | s, a)} \big[ r(s, a) + \gamma V(s') \big].$$

Therefore, from the offline dataset $\mathcal{D}$, we can define the following empirical Bellman operator:

$$\widehat{\mathcal{T}}_d^\mu V(s) := \begin{cases} \dfrac{1}{|\mathcal{D}_s|} \displaystyle\sum_{(s,a,r,s') \in \mathcal{D}_s} \big[ r + \gamma V(s') \big], & \text{if } |\mathcal{D}_s| > 0, \\ 0, & \text{otherwise,} \end{cases} \tag{2}$$

where $\mathcal{D}_s \subseteq \mathcal{D}$ is the set of transitions in the dataset $\mathcal{D}$ that begin in state $s$, $|\mathcal{D}_s|$ is the number of such transitions, and $\gamma$ is the discount factor.

We then show that the empirical Bellman operator has a unique fixed point.

**Proposition 4.1.** *The empirical Bellman operator $\widehat{\mathcal{T}}_d^\mu$ has a unique fixed point $V_d^\mu(s)$ such that*

$$\widehat{\mathcal{T}}_d^\mu V_d^\mu(s) = V_d^\mu(s). \tag{3}$$

From Proposition 4.1, we can learn a value function $V_d^\mu(s)$ from the offline dataset $\mathcal{D}$. Intuitively, when $\mathcal{D}$ is a nearly full-coverage offline dataset, the learned $V_d^\mu(s)$ provides an accurate approximation of the true value function $V^\mu(s)$, such that $V_d^\mu(s) \approx V^\mu(s)$.

We then train another approximated value function corresponding to the behavior policy with the help of a learned model. We learn an approximate dynamics model $P_\theta(s', r | s, a)$ through maximum likelihood estimation and define the Bellman operator associated with $P_\theta$ as follows:

$$\widehat{\mathcal{T}}_m^\mu V(s) := \mathbb{E}_{a \sim \mu(\cdot | s), (r, s') \sim P_\theta} \big[ r + \gamma V(s') \big]. \tag{4}$$

Note that when the learned model $P_\theta(s', r | s, a)$ provides an accurate approximation of the true environment model, denoted as $P^\star(s', r | s, a)$ (which jointly models the transition dynamics and reward), the induced Bellman operator $\widehat{\mathcal{T}}_m^\mu$ coincides with the true Bellman operator $\mathcal{T}^\mu$.

**Proposition 4.2.** *The Bellman operator induced by $P_\theta$, denoted $\widehat{\mathcal{T}}_m^\mu$, has a unique fixed point $V_m^\mu(s)$ such that*

$$\widehat{\mathcal{T}}_m^\mu V_m^\mu(s) = V_m^\mu(s). \tag{5}$$

From Proposition 4.2, we can iteratively apply $\widehat{\mathcal{T}}_m^\mu$ to obtain the unique value function $V_m^\mu$, representing the value function learned from the model. This value function serves as another approximation of $V^\mu(s)$, such that $V_m^\mu(s) \approx V^\mu(s)$.

The core idea is that if the learned model $P_\theta$ is accurate, the value function $V_m^\mu(s)$—obtained from the model—should align with $V_d^\mu(s)$—derived from the dataset—as both aim to approximate the true value function $V^\mu(s)$. Therefore, we define the value function inconsistency loss as follows:

$$\mathcal{L}_{vic}(\theta) = \mathbb{E}_{s \sim \rho_0} \Big[ \big( V_d^\mu(s) - V_m^\mu(s) \big)^2 \Big], \tag{6}$$

where $\rho_0$ denotes the initial distribution, and $\mathcal{L}_{vic}$ depends on $\theta$ because $V_m^\mu(s)$ is derived from the learned model $P_\theta$.

To learn the model, we begin with minimizing the negative log-likelihood, which we refer to as the original model (learning) loss,

$$\mathcal{L}_{\text{ori}}(\theta) = -\mathbb{E}_\mathcal{D} \big[ \log P_\theta(s', r | s, a) \big], \tag{7}$$

where $P_\theta$ denotes the parameterized probabilistic model within the hypothesis class $\mathcal{P}$ defined in Section 3.3.

We incorporate the value function inconsistency loss into the original model loss to construct the following augmented model loss:

$$\mathcal{L}_{aug}(\theta) = \mathcal{L}_{ori}(\theta) + \lambda \mathcal{L}_{vic}(\theta), \tag{8}$$

where $\lambda > 0$ is a user-chosen hyperparameter that balances the contributions of the two loss components. Thus, $\mathcal{L}_{aug}$

serves as the overall loss for training the dynamics model which aims to optimize both the original model loss and the value function inconsistency loss.

The conceptual foundation of value-aligned model training has demonstrated remarkable success in online environments, notably with MuZero (Schrittwieser et al., 2020). Distinct from MuZero's reliance on continuous online interaction and search-based policy iteration, VIPO is specifically designed for the offline regime, where it systematically mitigates model errors by penalizing the inconsistency between the value function derived directly from the static dataset and the one estimated through the learned dynamics.

### 4.2. Theoretical Foundation

To strictly justify the effectiveness of our approach, we analyze the model learning error of VIPO.

Let $\mathbb{P}^{\mu}_{P^{\star}}$ denote the trajectory distribution induced by $s_0 \sim \rho_0$, $a_t \sim \mu(\cdot \mid s_t)$, and $(r_t, s_{t+1}) \sim P^{\star}(\cdot \mid s_t, a_t)$. We define the normalized $\gamma$-discounted occupancy measure under $P^{\star}$ as: $d_{\mu}(s, a) := (1 - \gamma) \sum_{t=0}^{\infty} \gamma^t \mathbb{P}^{\mu}_{P^{\star}}(s_t = s, a_t = a)$.

**Assumption 4.3** (Realizability). The ground-truth joint environment model $P^{\star}(s', r \mid s, a)$ belongs to our model class $\mathcal{P} = \{P_{\theta}\}_{\theta \in \Theta}$; equivalently, there exists a parameter $\theta^{\star}$ in our model family such that $P_{\theta^{\star}} = P^{\star}$.

---

**Theorem 4.4** (Model learning error). *Let* $\mathcal{D} = \{(s_i, a_i, r_i, s'_i)\}_{i=1}^{N}$ *be an offline dataset of $N$ i.i.d. samples generated from $(s_i, a_i) \sim d_{\mu}$ and $(r_i, s'_i) \sim P^{\star}(\cdot \mid s_i, a_i)$. Let*

$$\theta^{\text{opt}} \in \arg\min_{\theta} \mathcal{L}_{\text{aug}}(\theta),$$

*and define the **model learning error** as*

$$\mathcal{E}(\theta^{\text{opt}}) := \mathbb{E}_{d_{\mu}} \left[ \mathrm{TV}\big(P_{\theta^{\text{opt}}}(\cdot \mid s, a), P^{\star}(\cdot \mid s, a)\big) \right],$$

*where* $\mathrm{TV}(\mu, \nu) := \sup_{A \subseteq \mathcal{X}} |\mu(A) - \nu(A)|$ *denotes the total variation distance between two probability measures $\mu$ and $\nu$ on $\mathcal{X}$.*

*Under Assumption 4.3, with probability at least $1 - \delta$, we have*

$$\mathcal{E}(\theta^{\text{opt}}) \leq \underbrace{\sqrt{\frac{c_1}{\delta}} \left( \frac{\log \mathcal{N}_{\mathcal{P}}(1/N)}{N} \right)^{1/4}}_{\text{Model generalization error}}$$
$$+ \underbrace{\sqrt{\frac{2\lambda c_1}{\delta}} \left( \frac{\log \mathcal{N}_{\mathcal{V}}(1/N)}{N} \right)^{1/4}}_{\text{Inconsistency generalization error}} \quad (9)$$
$$+ \underbrace{\sqrt{\frac{\lambda}{2} \mathcal{L}_{\text{vic}}(\theta^{\star})}}_{\text{Alignment bias}}.$$

---

Theorem 3.4 decomposes the model learning error into three interpretable components: model generalization error, inconsistency generalization error, and alignment bias. The former two terms quantify the statistical error arising from limited offline data, scaling with the complexities of the dynamics model hypothesis space and the value inconsistency loss class, respectively. As the dataset size $N \to \infty$, these generalization errors naturally converge to zero. The third term, $\mathcal{L}_{\text{vic}}(\theta^{\star})$, captures a finite-sample alignment bias: even when $P_{\theta^{\star}} = P^{\star}$, it can be non-zero because $V_d^{\mu}$ is an empirical estimate from $\mathcal{D}$ and may differ from the true value $V^{\mu}$, inducing residual mismatch. As $N \to \infty$, $V_d^{\mu} \to V^{\mu}$, so this bias disappears. Consequently, our analysis establishes the asymptotic consistency of VIPO, guaranteeing that the learned model converges to $P^{\star}$ in total variation distance as the sample size approaches infinity.

### 4.3. Model Gradient Theorem

To optimize $\mathcal{L}_{\text{aug}}$, it is essential to compute its gradient. However, in deriving the gradient of the augmented loss function, a significant challenge emerges from the implicit dependency of the model-learned value function $V_m^{\mu}(s)$ on the dynamics model parameters $\theta$. Unlike the original loss term $\mathcal{L}_{\text{ori}}$, whose gradient can be readily computed through automatic differentiation, the value function $V_m^{\mu}(s)$ is obtained via a recursive Bellman backup defined in Eq. (4) that involves multiple steps of state transitions and discounted future rewards. This recursive structure embeds a hidden dependence on $\theta$, thereby precluding the direct application of conventional differentiation techniques.

In the following, we establish the expression for the gradient of the augmented loss function, which forms the foundation for implementing VIPO. A key component of our approach is the recursive unrolling of gradients, allowing us to track the impact of model parameters across multiple transitions.

Let $\rho_{\theta}^{\mu}(s \to s', t)$ denote the state density at $s'$ after transitioning for $t$ time steps from state $s$ under the policy $\mu$ and the learned model $P_{\theta}$. Denote the (improperly) discounted state transition probability from $s$ to $s'$ by $d_{\theta}^{\mu}(s, s')$, defined as $d_{\theta}^{\mu}(s, s') = \sum_{t=0}^{\infty} \gamma^t \rho_{\theta}^{\mu}(s \to s', t)$.

---

**Theorem 4.5** (Model Gradient Theorem). *Let $\theta$ represent the parameters of the dynamics model $P_{\theta}$. Denote $V_d^{\mu}(\cdot)$ and $V_m^{\mu}(\cdot)$ as the value functions learned from the offline dataset $\mathcal{D}$, satisfying Eq. (3), and the model $P_{\theta}$, satisfying Eq. (5), respectively. Then:*

$$\nabla_{\theta} \mathcal{L}_{\text{aug}}(\theta) = \nabla_{\theta} \mathcal{L}_{\text{ori}}(\theta) - 2\lambda \mathbb{E}\Big[ \big(V_d^{\mu}(s) - V_{m,\theta}^{\mu}(s)\big)$$
$$\cdot \big(r' + \gamma V_{m,\theta}^{\mu}(s'')\big) \nabla_{\theta} \log P_{\theta}(s'', r' \mid s', a')\Big],$$

*where the expectation is taken over $s \sim \rho_0$, $s' \sim d_{\theta}^{\mu}$, $a' \sim \mu(\cdot \mid s')$, and $(s'', r') \sim P_{\theta}(\cdot \mid s', a')$.*

---

---

**Algorithm 1** VIPO: Value Function Inconsistency Penalized Offline Reinforcement Learning

---

1: **Require:** Offline dataset $\mathcal{D} = \{(s_i, a_i, r_i, s'_i)\}_{i=1}^N$; Initial model parameters $\theta$; Regularization coefficient $\lambda$; Learning rate $\eta$; Maximum number of iterations $T$.
2: Set iteration counter $t \leftarrow 0$.
3: **while** not reaching maximum iterations $T$ or convergence criterion **do**
4:     Compute original model loss via Eq. (7).
5:     Use the offline dataset $\mathcal{D}$ to compute the value function $V_d(s)$ by minimizing Eq. (10).
6:     Use the current model $P_\theta$ to compute the model value function $V_m(s)$ by minimizing Eq. (11).
7:     Update $\theta$ using gradient descent via Eq. (12).
8:     Update the iteration counter: $t \leftarrow t + 1$.
9:     Check convergence criterion; if satisfied, exit the loop.
10: **end while**
11: Obtain the optimized model parameters $\theta^* \leftarrow \theta$. Let $\widehat{\mathcal{M}}$ be the MDP with learned dynamics model $P_{\theta^*}$.
12: (OPTIONAL) Use a behavior cloning approach to estimate the behavior policy $\mu$.
13: Run any RL algorithm on $\widehat{\mathcal{M}}$ until convergence to obtain $\pi_{\text{out}} \leftarrow \text{PLANNER}(\widehat{\mathcal{M}}, \pi_{\text{int}} = \mu)$.

---

Theorem 4.5 provides the gradient of the augmented model loss, which underlies the practical implementation of VIPO.

### 4.4. Practical Algorithm

We now introduce our practical algorithm, VIPO.

While our theoretical framework allows for general function approximation, in our practical implementation, we parameterize $P_\theta$ by a neural network that models the next state and reward as a Gaussian distribution conditioned on the current state and action, i.e., $P_\theta(s'_t, r_t \mid s_t, a_t) = \mathcal{N}(\mu_\theta(s_t, a_t), \Sigma_\theta(s_t, a_t))$.

In VIPO, we also employ neural networks to approximate the value function. To approximate $V_d^\mu(s)$, we learn $V_d(s)$ by minimizing the following empirical mean squared Bellman error (MSBE):

$$\mathcal{L}_{V_d}(\varphi_d) = \mathbb{E}_{(s,a,r,s')\sim\mathcal{D}}\left[\left(r + \gamma \bar{V}_d(s') - V_d(s)\right)^2\right], \quad (10)$$

where $\varphi_d$ is the parameter of primary state value network $V_d$ and $\bar{V}_d$ is the target state value network. $\bar{V}_d$ is obtained using an exponentially moving average of parameters of the state value network (soft update): $\bar{\varphi}_d \leftarrow \tau \varphi_d + (1-\tau)\bar{\varphi}_d$, where $\tau \in [0, 1]$. Eq. (10) serves as the surrogate objective for learning the Bellman backup defined in Eq. (2).

We then learn $V_m(s)$ to approximate $V_m^\mu(s)$ by minimizing the following loss function:

$$\mathcal{L}_{V_m}(\varphi_m) = \mathbb{E}_{(s,a)\sim\mathcal{D}, (r,s')\sim P_\theta}\left[\left(r + \gamma \bar{V}_m(s') - V_m(s)\right)^2\right], \quad (11)$$

where $\varphi_m$ is the parameter of the primary network $V_m$, and $\bar{V}_m$ is the target network obtained by the exponentially

moving average of parameters of $V_m$. Note that $r$ and $s'$ are sampled from the learned dynamic model $P_\theta$ instead of from the offline dataset. Eq. (11) serves as the surrogate for learning the Bellman backup defined in Eq. (4).

According to Theorem 4.5, calculating the exact gradient requires computing an expectation over the state-action visitation distribution $d_\theta^\mu$ induced by the learned model. However, sampling from this distribution online is computationally expensive and potentially unstable. Instead, we adopt a standard off-policy approximation by replacing the model-induced distribution $d_\theta^\mu$ with the empirical data distribution $\mathcal{D}$. Specifically, we sample the tuple $(s, a)$ directly from the offline dataset to compute the gradient update.

This enables effective training using solely the offline dataset, yielding the following surrogate gradient:

$$\nabla_\theta \mathcal{L}_{aug}^{surr}(\theta) = \nabla_\theta \mathcal{L}_{ori}(\theta) - 2\lambda \mathbb{E}_{(s,a)\sim\mathcal{D},(s',r)\sim P_\theta}\left[(\bar{V}_d(s) \right.$$
$$\left. - \bar{V}_m(s))(r + \gamma \bar{V}_m(s'))\nabla_\theta \log P_\theta(s', r|s, a)\right]. \quad (12)$$

> **Remark.** We next provide intuition for why the proposed surrogate gradient is effective by drawing an analogy to the policy gradient theorem (Sutton et al., 1999): $\nabla_\theta J(\theta) = \mathbb{E}_{s\sim\rho^{\pi_\theta}, a\sim\pi_\theta}[Q^{\pi_\theta}(s, a)\nabla_\theta \log \pi_\theta(a|s)]$. In our surrogate gradient, the term $r + \gamma \bar{V}_m(s')$ serves as an approximation of the Q-value corresponding to the learned model. Analogous to the policy gradient case, where it updates the policy $\pi_\theta$ to maximize $J(\theta)$, in our model gradient case, $(r+\gamma \bar{V}_m(s'))\nabla_\theta \log P_\theta(s', r|s, a)$ updates $P_\theta$ to improve the value function. Specifically, $(r + \gamma \bar{V}_m(s'))\nabla_\theta \log P_\theta(s', r|s, a)$ is the direction that increases $V_m$. Multiplying this term by $-2\lambda(\bar{V}_d(s) - \bar{V}_m(s))$ ensures that when $\bar{V}_d(s) > \bar{V}_m(s)$, the gradient descent update in our algorithm updates the model to increase $\bar{V}_m$ towards $\bar{V}_d$, and vice versa. Consequently, this surrogate gradient promotes consistency between value functions.

We now present the complete algorithm flow in Algorithm 1, which outlines the dynamics model training process of VIPO. The process begins by computing the original model loss using Eq. (7). Next, $V_d(s)$ and $V_m(s)$ are learned by minimizing Eq. (10) and Eq. (11), respectively. With these components in place, the model parameters $\theta$ are updated using Eq. (12).

## 5. Experiments

In this section, we empirically demonstrate that VIPO consistently learns a highly accurate model and outperforms previous methods through three distinct perspectives: Benchmark Results (Subsection 5.1), Uncertainty Analysis (Subsection 5.2), and Predictive Capability (Subsection 5.3).

*Table 1.* Normalized average returns on D4RL Gym tasks. The experiments are run on MuJoCo-"v2" datasets over 4 random seeds. r = random, m = medium, m-r = medium-replay, m-e = medium-expert. MOPO* indicates the results of MOPO retrained in "v2" datasets as reported by Sun et al. (2023). The returns labeled with * in random tasks indicate values obtained from our training, as they were not reported in the original paper. We **bold** the highest mean.

| Task Name | TD3+BC | CQL | IQL | DQL | MOPO* | MOReL | COMBO | RAMBO | MOBILE | VIPO-MOPO | VIPO-MOBILE |
|---|---|---|---|---|---|---|---|---|---|---|---|
| *halfcheetah-r* | 10.2 | 31.3 | 16.1* | 22.1* | 38.5 | 25.6 | 38.8 | 39.5 | 39.3 | $38.9_{\pm0.7}$ | $\mathbf{42.5_{\pm0.2}}$ |
| *hopper-r* | 11.0 | 5.3 | 10.8* | 17.5* | 31.7 | **53.6** | 17.9 | 25.4 | 31.9 | $30.2_{\pm3.2}$ | $33.4_{\pm1.9}$ |
| *walker2d-r* | 1.4 | 5.4 | 7.7* | 14.1* | 7.4 | **37.3** | 7.0 | 0.0 | 17.9 | $9.5_{\pm0.8}$ | $20.0_{\pm0.1}$ |
| *halfcheetah-m* | 42.8 | 46.9 | 47.4 | 51.1 | 73.0 | 42.1 | 54.2 | 77.9 | 74.6 | $78.6_{\pm2.1}$ | $\mathbf{80.0_{\pm0.4}}$ |
| *hopper-m* | 99.5 | 61.9 | 66.3 | 90.5 | 62.8 | 95.4 | 97.2 | 87.0 | 106.6 | $73.5_{\pm1.7}$ | $\mathbf{107.7_{\pm1.0}}$ |
| *walker2d-m* | 79.7 | 79.5 | 78.3 | 87.0 | 84.1 | 77.8 | 81.9 | 84.9 | 87.7 | $87.9_{\pm2.9}$ | $\mathbf{93.1_{\pm1.8}}$ |
| *halfcheetah-m-r* | 43.4 | 45.3 | 44.2 | 47.8 | 72.1 | 40.2 | 55.1 | 68.7 | 71.7 | $73.0_{\pm1.8}$ | $\mathbf{77.2_{\pm0.4}}$ |
| *hopper-m-r* | 31.4 | 86.3 | 94.7 | 101.3 | 103.5 | 93.6 | 89.5 | 99.5 | 103.9 | $103.0_{\pm0.7}$ | $\mathbf{109.6_{\pm0.9}}$ |
| *walker2d-m-r* | 25.2 | 76.8 | 73.9 | 95.5 | 85.6 | 49.8 | 56.0 | 89.2 | 89.9 | $88.2_{\pm0.5}$ | $\mathbf{98.4_{\pm0.3}}$ |
| *halfcheetah-m-e* | 97.9 | 95.0 | 86.7 | 96.8 | 90.8 | 53.3 | 90.0 | 95.4 | 108.2 | $93.8_{\pm2.7}$ | $\mathbf{110.0_{\pm0.4}}$ |
| *hopper-m-e* | 112.2 | 96.9 | 91.5 | 111.1 | 81.6 | 108.7 | 111.1 | 88.2 | 112.6 | $104.5_{\pm3.1}$ | $\mathbf{113.2_{\pm0.1}}$ |
| *walker2d-m-e* | 105.7 | 109.1 | 109.6 | 110.1 | 112.9 | 95.6 | 103.3 | 56.7 | 115.2 | $111.3_{\pm1.0}$ | $\mathbf{117.7_{\pm1.0}}$ |
| **Average** | 54.6 | 61.6 | 60.6 | 70.4 | 70.3 | 64.4 | 66.8 | 67.7 | 80.0 | 74.4 | **83.6** |

## 5.1. Benchmark Results

In Subsection 5.1, we evaluate VIPO's performance on the D4RL, NeoRL, and the challenging AntMaze benchmarks. Specifically, we replace the models used in previous methods (MOPO (Yu et al., 2020), MOBILE (Sun et al., 2023), and LEQ (Park & Lee, 2024)) with the model trained by VIPO, while retaining their policy extraction components (i.e., the planner) to enable a *fair comparison*. This substitution results in significant performance improvements, enabling VIPO-MOBILE (an integration over the prior SOTA method MOBILE) to achieve new SOTA performance on most tasks. We emphasize that the representative methods MOPO, MOBILE, and LEQ (specifically for the AntMaze benchmarks in Table 3) are selected merely as examples. In principle, VIPO can be integrated with *any existing model-based offline RL algorithm*, and such integration is expected to yield performance improvements.

### 5.1.1. D4RL EXPERIMENT RESULTS

The D4RL experimental results are summarized in Table 1, reporting the normalized average scores obtained from the final online evaluation. We compare VIPO against nine representative offline RL algorithms, comprising four model-free methods (TD3+BC (Fujimoto & Gu, 2021), CQL (Kumar et al., 2020), IQL (Kostrikov et al., 2021), DQL (Wang et al., 2022)) and five model-based methods (MOPO (Yu et al., 2020), MOReL (Kidambi et al., 2020), COMBO (Yu et al., 2021), RAMBO (Rigter et al., 2022), MOBILE (Sun et al., 2023)). Note that the results for MOPO are labeled as MOPO*, sourced from Sun et al. (2023), as they were retrained on the 'v2' datasets to ensure a fair comparison (the original paper used 'v0'). For detailed descriptions of these baselines and experimental setups, please refer to Section E.1.

*Table 2.* Normalized average returns on NeoRL tasks. The experiments are run on MuJoCo-"v3" datasets over 4 random seeds. L = low, M = medium, H = high. We **bold** the highest mean.

| Task Name | CQL | TD3+BC | MOPO | MOBILE | VIPO-MOBILE |
|---|---|---|---|---|---|
| *halfcheetah-L* | 38.2 | 30.0 | 40.1 | 54.7 | $\mathbf{58.5 \pm 0.1}$ |
| *hopper-L* | 16.0 | 15.8 | 6.2 | 17.4 | $\mathbf{30.7 \pm 0.3}$ |
| *walker2d-L* | 44.7 | 43.0 | 11.6 | 37.6 | $\mathbf{67.6 \pm 0.7}$ |
| *halfcheetah-M* | 54.6 | 52.3 | 62.3 | 77.8 | $\mathbf{80.9 \pm 0.2}$ |
| *hopper-M* | 64.5 | **70.3** | 1.0 | 51.1 | $66.3 \pm 0.2$ |
| *walker2d-M* | 57.3 | 58.5 | 39.9 | 62.2 | $\mathbf{76.8 \pm 0.1}$ |
| *halfcheetah-H* | 77.4 | 75.3 | 65.9 | 83.0 | $\mathbf{89.4 \pm 0.6}$ |
| *hopper-H* | 76.6 | 75.3 | 11.5 | 87.8 | $\mathbf{107.7 \pm 0.5}$ |
| *walker2d-H* | 75.3 | 69.6 | 18.0 | 74.9 | $\mathbf{81.7 \pm 1.0}$ |
| **Average** | 56.1 | 54.5 | 28.5 | 60.7 | **73.3** |

Our results indicate that VIPO-MOBILE (an integration with the prior SOTA method MOBILE) outperforms all baseline methods on most tasks, achieving the highest average score compared to the other approaches. Since we only replace the dynamics model and keep the planner unchanged, the notable improvements observed with VIPO-MOPO over MOPO and VIPO-MOBILE over MOBILE provide empirical evidence that the value inconsistency approach results in more effective models compared to those trained with the original model loss function.

### 5.1.2. NEORL EXPERIMENT RESULTS

In this experiment, we selected four offline RL algorithms as baselines: CQL, TD3+BC, MOPO, and MOBILE. The experimental results are summarized in Table 2, where (L, M, H) denote three dataset types—low, medium, and high quality, respectively. Notably, VIPO-MOBILE surpasses the previous state-of-the-art method MOBILE by an average margin of over **20%**, particularly across the HalfCheetah, Hopper, and Walker tasks. This outstanding performance on the challenging NeoRL benchmark highlights the potential of our method for real-world applications.

*Table 3.* Normalized average returns on AntMaze tasks. Each score is the average success rate over 10 trials. We **bold** the highest mean.

| Dataset | MOBILE | IQL-TD-MPC | LEQ | VIPO-LEQ |
|---|---|---|---|---|
| *umaze* | 0.0 | 52.0 | 94.4 ± 6.3 | **95.7 ± 4.9** |
| *umaze-diverse* | 0.0 | 72.6 | 71.0 ± 12.3 | **74.8 ± 12.7** |
| *large-play* | 0.0 | 66.6 | 58.6 ± 9.1 | **79.5 ± 4.8** |
| *large-diverse* | 0.0 | 4.0 | 60.2 ± 18.3 | **81.8 ± 8.3** |
| *ultra-play* | 0.0 | 20.6 | 25.8 ± 18.2 | **31.0 ± 18.7** |
| *ultra-diverse* | 0.0 | 3.6 | **55.8 ± 18.3** | 26.5 ± 10.9 |
| **Average** | 0 | 36.6 | 61.0 | **64.9** |

### 5.1.3. ANTMAZE EXPERIMENT RESULTS

Model-based offline RL methods often struggle in long-horizon, sparse-reward settings such as AntMaze, where learning an accurate dynamics model from static data is difficult and small model errors can compound during planning. To demonstrate VIPO's robustness in these challenging environments, we compare it with three model-based baselines (MOBILE, IQL-TD-MPC (Chitnis et al., 2024), and LEQ (Park & Lee, 2024)) on six AntMaze tasks (Table 3). The `large` and `ultra` variants are widely regarded as particularly challenging benchmarks. Notably, the prior SOTA method MOBILE fails completely on these tasks, achieving a score of 0. However, by substituting the LEQ model with our VIPO dynamics model, we observe performance improvements across the majority of tasks compared to the original LEQ method. These results highlight the advantage of our approach in learning an accurate model to enhance long-horizon planning under sparse rewards.

### 5.2. Revisiting Uncertainty

In this Subsection, we design an experiment to reevaluate the uncertainty defined in MOPO (Yu et al., 2020). Specifically, we remove a portion of the Walker2d-medium-replay dataset and train models on the reduced dataset using both MOPO and VIPO. Intuitively, a higher drop ratio, indicating less available data about the environment, should result in higher model uncertainty. However, models trained with MOPO fail to capture this trend while models trained with VIPO exhibit a clear positive correlation.

### 5.2.1. UNCERTAINTY DEFINITION

In MOPO, the max-aleatoric uncertainty quantifier is utilized, with the model dynamics represented as $\hat{P}_k(s'|s, a) = \mathcal{N}(\mu_k(s, a), \Sigma_k(s, a))$, where $\mu_k$ and $\Sigma_k$ denote the mean and covariance matrix of the multivariate Gaussian modeling the $k$-th transition dynamics in the ensemble. The uncertainty is defined as

$$U(s, a) := \max_k \|\Sigma_k(s, a)\|_{\mathrm{F}}, \quad (13)$$

where $\|\cdot\|_{\mathrm{F}}$ denotes the Frobenius norm. As justified in MOPO, this uncertainty estimator captures both epistemic

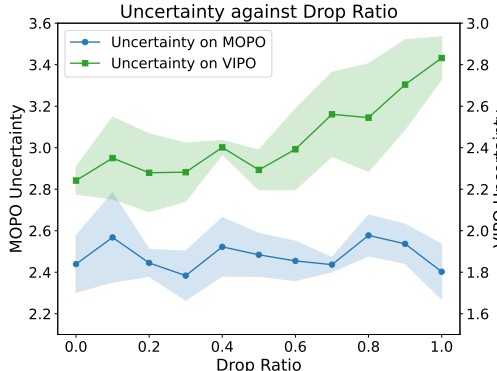

*Figure 2.* Uncertainty of models trained by MOPO and VIPO, averaged over 4 random seeds.

and aleatoric uncertainty of the true dynamics. To ensure a fair and consistent comparison, we adopt the same estimator to quantify the uncertainty of the models learned by both MOPO and VIPO.

### 5.2.2. EXPERIMENT SETUP

The experiment is conducted on the D4RL Walker2d-medium-replay dataset, which contains 301,698 transitions $(s, a, r, s')$. From this dataset, 1,000 transitions are randomly selected. For each selected $(s, a)$ pair, the dataset is searched to identify all other $(s', a')$ pairs that fall within the range $(s \pm 0.8, a \pm 0.8)$. These additional points, together with the original 1,000 points, form a candidate dataset, resulting in a total of 49,952 points. This approach allows us to capture data points with similar (s, a) values, which likely share similar contexts in the MDP.

To evaluate the robustness of uncertainty measures, dropout ratios ranging from 0% to 100% are applied to the candidate dataset in 10% increments. For each dropout ratio, a subset of points from the candidate dataset is randomly removed, and the remaining points are merged back into the original dataset to create modified datasets. An ensemble of transition models is trained on each modified dataset by MOPO and VIPO, respectively, and the uncertainty in the regions corresponding to the perturbed $(s, a)$ pairs is evaluated separately. The uncertainty is calculated using Eq. (13) and averaged across the selected 1000 state-action pairs.

### 5.2.3. EXPERIMENT RESULTS

Our experiment is based on the premise that decreasing the amount of data should lead to higher uncertainty in a well-learned model, because limited information about the environment naturally entails greater uncertainty. To test this, we progressively drop portions of the candidate dataset and train models with MOPO and VIPO under identical settings. We use Eq. (13) to evaluate the uncertainties. As illustrated in Figure 2, VIPO's uncertainty grows consistently with the drop ratio, aligning with the expected trend,

*Table 4.* Model predictive error comparison on D4RL Gym tasks, averaged over 6 random seeds.

| Task Name | OL Model Error | VIPO Model Error |
|---|---|---|
| *halfcheetah-r* | $0.079 \pm 0.008$ | $0.073 \pm 0.002$ |
| *hopper-r* | $0.0003 \pm 1e^{-4}$ | $0.0003 \pm 1e^{-4}$ |
| *walker2d-r* | $0.295 \pm 0.024$ | $0.247 \pm 0.013$ |
| *halfcheetah-m* | $0.550 \pm 0.060$ | $0.193 \pm 0.035$ |
| *hopper-m* | $0.009 \pm 0.002$ | $0.007 \pm 3e^{-4}$ |
| *walker2d-m* | $0.287 \pm 0.122$ | $0.270 \pm 0.085$ |
| *halfcheetah-m-r* | $0.396 \pm 0.045$ | $0.289 \pm 0.023$ |
| *hopper-m-r* | $0.017 \pm 0.006$ | $0.014 \pm 0.002$ |
| *walker2d-m-r* | $0.285 \pm 0.020$ | $0.219 \pm 0.005$ |
| *halfcheetah-m-e* | $0.081 \pm 0.024$ | $0.070 \pm 0.010$ |
| *hopper-m-e* | $0.002 \pm 6e^{-4}$ | $0.002 \pm 4e^{-4}$ |
| *walker2d-m-e* | $0.077 \pm 0.002$ | $0.070 \pm 0.0026$ |
| **Average** | $0.173 \pm 0.026$ | $\mathbf{0.121 \pm 0.015}$ |

whereas MOPO's uncertainty remains largely insensitive to data reduction. This result highlights VIPO's potential to capture uncertainty more faithfully and to learn a more accurate model.

### 5.3. Predictive Capability

In this subsection, we evaluate the predictive capability of the model trained with VIPO compared to a model trained using the original model loss, which does not account for value function inconsistency—the latter has been adopted by MOPO, MOReL, and MOBILE. The results on the D4RL benchmark indicate that VIPO achieves lower predictive error.

In this experiment, we train a model using only the original loss (referred to as OL Model in Table 4) defined in Eq. (7), which aligns with the approach adopted by previous methods such as MOPO, MOReL, and MOBILE. Subsequently, we train a model with VIPO. Finally, we assess the predictive accuracy of the OL Model and VIPO Model by comparing the predicted reward and next state $(\hat{r}, \hat{s}')$ with the true values $(r, s')$ for a given state-action pair $(s, a)$.

For all tasks, the dataset is split into training and validation sets in a 9:1 ratio. The model predictive error is measured as the mean squared error between the predicted and true values on the validation set. To ensure consistency, the same validation set is used for evaluating model performance across all tasks.

As summarized in Table 4, our results demonstrate that the VIPO achieves lower predictive error than the OL Model on the majority of tasks. On average, VIPO reduces the model error from $0.173 \pm 0.026$ (OL Model) to $0.121 \pm 0.015$, reflecting a relative reduction of approximately **30%**. This validates the effectiveness of incorporating the value function inconsistency loss into model training. These findings

further confirm that VIPO can effectively learn a highly accurate model.

## 6. Conclusion

This paper proposes VIPO, a model-based offline RL algorithm that incorporates value function inconsistency into model training. Through extensive evaluations from multiple perspectives, we show that VIPO learns accurate models efficiently and consistently outperforms existing methods. As a general and readily applicable framework, VIPO can be integrated into model-based offline RL methods to improve model accuracy. Our work may encourage researchers to exploit data more comprehensively, rather than relying solely on costly architectures such as diffusion or transformers.

## Acknowledgements

This work was supported by the Singapore Ministry of Education Tier 2 Academic Research Funds (T2EP20123-0037, T2EP20224-0035).

## Impact Statement

This paper presents work whose goal is to advance the field of Machine Learning. There are many potential societal consequences of our work, none of which we feel must be specifically highlighted here.

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

# Appendix

## Table of Contents

## A. Proof of the Theorems

**Proof of Theorem 4.4**

*Proof.* We first introduce the population risks associated with the empirical losses $\mathcal{L}_{\mathrm{ori}}$ and $\mathcal{L}_{\mathrm{vic}}$:

$$\mathcal{L}_{\mathrm{ori}}^{\mathrm{pop}}(\theta) := \mathbb{E}_{(s,a)\sim d_\mu} \mathbb{E}_{(s',r)\sim P^\star(\cdot|s,a)}\Big[ -\log P_\theta(s', r \mid s, a)\Big], \tag{14}$$

$$\mathcal{L}_{\mathrm{vic}}^{\mathrm{pop}}(\theta) := \mathbb{E}_{s\sim d_\mu}\Big[\big(V_d^\mu(s) - V_{m,\theta}^\mu(s)\big)^2\Big]. \tag{15}$$

Let $Z := (s, a, r, s')$ and let $\mathbb{P}$ denote the joint distribution of $Z$ induced by the data model, namely $(s, a) \sim d_\mu$ and $(r, s') \sim P^\star(\cdot \mid s, a)$. Let $\mathbb{P}_N$ be the empirical measure induced by the dataset $\mathcal{D} = \{Z_i\}_{i=1}^N$. Consider the loss classes

$$\mathcal{F}_{\mathrm{ori}} := \{-\log P_\theta(s', r \mid s, a) : \theta \in \Theta\}, \qquad \mathcal{F}_{\mathrm{vic}} := \{(V_d^\mu(s) - V_{m,\theta}^\mu(s))^2 : \theta \in \Theta\}.$$

By the bracketing maximal inequality for empirical processes under $L_2(\mathbb{P})$ bracketing entropy control (Van Der Vaart & Wellner, 1996; Geer, 2000), there exists an absolute constant $C > 0$ such that

$$\mathbb{E}_{\mathcal{D}}\Big[ \sup_\theta \big|\mathcal{L}_{\mathrm{ori}}(\theta) - \mathcal{L}_{\mathrm{ori}}^{\mathrm{pop}}(\theta)\big|\Big] = \mathbb{E}_{\mathcal{D}}\Big[ \sup_{f\in\mathcal{F}_{\mathrm{ori}}} \big|(\mathbb{P}_N - \mathbb{P})f\big|\Big] \leq C\sqrt{\frac{\log\mathcal{N}_\mathcal{P}(1/N)}{N}}, \tag{16}$$

$$\mathbb{E}_{\mathcal{D}}\Big[ \sup_\theta \big|\mathcal{L}_{\mathrm{vic}}(\theta) - \mathcal{L}_{\mathrm{vic}}^{\mathrm{pop}}(\theta)\big|\Big] = \mathbb{E}_{\mathcal{D}}\Big[ \sup_{g\in\mathcal{F}_{\mathrm{vic}}} \big|(\mathbb{P}_N - \mathbb{P})g\big|\Big] \leq C\sqrt{\frac{\log\mathcal{N}_\mathcal{V}(1/N)}{N}}. \tag{17}$$

Here $\mathbb{E}_{\mathcal{D}}$ denotes expectation with respect to the randomness of the dataset $\mathcal{D} = \{Z_i\}_{i=1}^N$, where $Z_i = (s_i, a_i, r_i, s_i') \overset{\mathrm{i.i.d.}}{\sim} \mathbb{P}$.

Applying Markov's inequality to Eq. (16)–Eq. (17) with confidence level $\delta/2$ and then taking a union bound, there exists an

absolute constant $c_1 > 0$ such that, with probability at least $1 - \delta$,

$$\sup_\theta \left| \mathcal{L}_{\text{ori}}(\theta) - \mathcal{L}_{\text{ori}}^{\text{pop}}(\theta) \right| \leq \Delta_\mathcal{P} := \frac{c_1}{\delta} \sqrt{\frac{\log \mathcal{N}_\mathcal{P}(1/N)}{N}}, \tag{18}$$

$$\sup_\theta \left| \mathcal{L}_{\text{vic}}(\theta) - \mathcal{L}_{\text{vic}}^{\text{pop}}(\theta) \right| \leq \Delta_\mathcal{V} := \frac{c_1}{\delta} \sqrt{\frac{\log \mathcal{N}_\mathcal{V}(1/N)}{N}}. \tag{19}$$

In the following, we work on the event where Eq. (18)–Eq. (19) hold. Since $\theta^{\text{opt}} \in \arg\min_\theta \mathcal{L}_{\text{aug}}(\theta)$,

$$\mathcal{L}_{\text{ori}}(\theta^{\text{opt}}) + \lambda \mathcal{L}_{\text{vic}}(\theta^{\text{opt}}) \leq \mathcal{L}_{\text{ori}}(\theta^\star) + \lambda \mathcal{L}_{\text{vic}}(\theta^\star).$$

Using $\mathcal{L}_{\text{vic}}(\theta^{\text{opt}}) \geq 0$ yields

$$\mathcal{L}_{\text{ori}}(\theta^{\text{opt}}) \leq \mathcal{L}_{\text{ori}}(\theta^\star) + \lambda \mathcal{L}_{\text{vic}}(\theta^\star).$$

Converting empirical losses to population risks via Eq. (18)–Eq. (19), we get

$$\begin{aligned}
\mathcal{L}_{\text{ori}}^{\text{pop}}(\theta^{\text{opt}}) &\leq \mathcal{L}_{\text{ori}}(\theta^{\text{opt}}) + \Delta_\mathcal{P} \\
&\leq \mathcal{L}_{\text{ori}}(\theta^\star) + \lambda \mathcal{L}_{\text{vic}}(\theta^\star) + \Delta_\mathcal{P} \\
&\leq \mathcal{L}_{\text{ori}}^{\text{pop}}(\theta^\star) + \Delta_\mathcal{P} + \lambda \big( \mathcal{L}_{\text{vic}}^{\text{pop}}(\theta^\star) + \Delta_\mathcal{V} \big) + \Delta_\mathcal{P},
\end{aligned}$$

and hence

$$\mathcal{L}_{\text{ori}}^{\text{pop}}(\theta^{\text{opt}}) - \mathcal{L}_{\text{ori}}^{\text{pop}}(\theta^\star) \leq 2\Delta_\mathcal{P} + \lambda \Delta_\mathcal{V} + \lambda \mathcal{L}_{\text{vic}}^{\text{pop}}(\theta^\star). \tag{20}$$

By Eq. (14) and the identity $\mathbb{E}_P[-\log Q] - \mathbb{E}_P[-\log P] = D_{\text{KL}}(P\|Q)$ applied conditionally on $(s,a)$, it holds that

$$\mathcal{L}_{\text{ori}}^{\text{pop}}(\theta^{\text{opt}}) - \mathcal{L}_{\text{ori}}^{\text{pop}}(\theta^\star) = \mathbb{E}_{(s,a) \sim d_\mu} \big[ D_{\text{KL}} \big( P^\star(\cdot \mid s, a) \,\|\, P_{\theta^{\text{opt}}}(\cdot \mid s, a) \big) \big].$$

From Pinsker's inequality, we have

$$\text{TV}\big(P^\star(\cdot \mid s, a), P_{\theta^{\text{opt}}}(\cdot \mid s, a)\big) \leq \sqrt{\tfrac{1}{2} D_{\text{KL}}\big(P^\star(\cdot \mid s, a) \,\|\, P_{\theta^{\text{opt}}}(\cdot \mid s, a)\big)}.$$

Taking expectation over $(s,a) \sim d_\mu$ and applying Jensen yields

$$\mathcal{E}(\theta^{\text{opt}}) \leq \sqrt{\tfrac{1}{2} \big( \mathcal{L}_{\text{ori}}^{\text{pop}}(\theta^{\text{opt}}) - \mathcal{L}_{\text{ori}}^{\text{pop}}(\theta^\star) \big)}.$$

Combining with Eq. (19) and using $\sqrt{x + y + z} \leq \sqrt{x} + \sqrt{y} + \sqrt{z}$, we obtain

$$\mathcal{E}(\theta^{\text{opt}}) \leq \sqrt{\Delta_\mathcal{P}} + \sqrt{\frac{\lambda}{2} \Delta_\mathcal{V}} + \sqrt{\frac{\lambda}{2} \mathcal{L}_{\text{vic}}^{\text{pop}}(\theta^\star)}. \tag{21}$$

By Eq. (18), we have

$$\mathcal{L}_{\text{vic}}^{\text{pop}}(\theta^\star) \leq \mathcal{L}_{\text{vic}}(\theta^\star) + \Delta_\mathcal{V}.$$

Therefore, using $\sqrt{u + v} \leq \sqrt{u} + \sqrt{v}$,

$$\sqrt{\frac{\lambda}{2} \mathcal{L}_{\text{vic}}^{\text{pop}}(\theta^\star)} \leq \sqrt{\frac{\lambda}{2} \mathcal{L}_{\text{vic}}(\theta^\star)} + \sqrt{\frac{\lambda}{2} \Delta_\mathcal{V}}.$$

Plugging this into Eq. (21) yields

$$\begin{aligned}
\mathcal{E}(\theta^{\text{opt}}) &\leq \sqrt{\Delta_\mathcal{P}} + 2\sqrt{\frac{\lambda}{2} \Delta_\mathcal{V}} + \sqrt{\frac{\lambda}{2} \mathcal{L}_{\text{vic}}(\theta^\star)} \\
&= \sqrt{\Delta_\mathcal{P}} + \sqrt{2\lambda \, \Delta_\mathcal{V}} + \sqrt{\frac{\lambda}{2} \mathcal{L}_{\text{vic}}(\theta^\star)}. \tag{22}
\end{aligned}$$

Finally, substituting the definitions of $\Delta_{\mathcal{P}}$ and $\Delta_{\mathcal{V}}$ from Eq. (17)–Eq. (18),

$$\Delta_{\mathcal{P}} = \frac{c_1}{\delta} \sqrt{\frac{\log \mathcal{N}_{\mathcal{P}}(1/N)}{N}}, \qquad \Delta_{\mathcal{V}} = \frac{c_1}{\delta} \sqrt{\frac{\log \mathcal{N}_{\mathcal{V}}(1/N)}{N}},$$

we have

$$\sqrt{\Delta_{\mathcal{P}}} = \sqrt{\frac{c_1}{\delta}} \left( \frac{\log \mathcal{N}_{\mathcal{P}}(1/N)}{N} \right)^{1/4}, \qquad \sqrt{2\lambda\,\Delta_{\mathcal{V}}} = \sqrt{\frac{2\lambda c_1}{\delta}} \left( \frac{\log \mathcal{N}_{\mathcal{V}}(1/N)}{N} \right)^{1/4}.$$

Hence,

$$\mathcal{E}(\theta^{\mathrm{opt}}) \le \sqrt{\frac{c_1}{\delta}} \left( \frac{\log \mathcal{N}_{\mathcal{P}}(1/N)}{N} \right)^{1/4} + \sqrt{\frac{2\lambda c_1}{\delta}} \left( \frac{\log \mathcal{N}_{\mathcal{V}}(1/N)}{N} \right)^{1/4} + \sqrt{\frac{\lambda}{2} \mathcal{L}_{\mathrm{vic}}(\theta^{\star})}.$$

$\square$

**Proof of Theorem 4.5**

*Proof.* The gradient of the original loss can be obtained through automatic differentiation. Now we focus on the gradient derivation of the value consistency loss as defined in Eq. (6).

$$\nabla_{\theta} \mathbb{E}_{s \sim \rho_0} \left[ \left( V_d^{\mu}(s) - V_m^{\mu}(s) \right)^2 \right] = -2\mathbb{E}_{s \sim \rho_0} \left[ \left( V_d^{\mu}(s) - V_m^{\mu}(s) \right) \nabla_{\theta} V_m^{\mu}(s) \right]. \tag{23}$$

Then, we need to calculate the gradient of $V_m^{\mu}(s)$ in terms of $\theta$. To achieve this goal, we start by decomposing the value function using the Bellman equation, which is inspired by the previous work (Rigter et al., 2022). For the given state $s$:

$$\begin{aligned} V_m^{\mu}(s) &= \sum_a \mu(a \mid s) Q_m(s, a) \\ &= \sum_a \mu(a \mid s) \sum_{s', r \sim P_{\theta}} \left( r + \gamma V_m^{\mu}(s') \right) \cdot P_{\theta}(s', r \mid s, a), \end{aligned} \tag{24}$$

where $Q_m(s, a)$ is the state-action value function under dynamics model $P_{\theta}$ and the behavior policy $\mu$. Applying the product rule:

$$\begin{aligned} \nabla_{\theta} V_m^{\mu}(s) &= \sum_a \mu(a \mid s) \sum_{s', r} \Big[ \left( r + \gamma V_m^{\mu}(s') \right) \cdot \nabla_{\theta} P_{\theta}(s', r \mid s, a) \\ &\quad + P_{\theta}(s', r \mid s, a) \cdot \nabla_{\theta} \left( r + \gamma V_m^{\mu}(s') \right) \Big] \\ &= \sum_a \mu(a \mid s) \sum_{s', r} \left( r + \gamma V_m^{\mu}(s') \right) \cdot \nabla_{\theta} P_{\theta}(s', r \mid s, a) \\ &\quad + \gamma \sum_a \mu(a \mid s) \sum_{s'} P_{\theta}(s' \mid s, a) \cdot \nabla_{\theta} V_m^{\mu}(s'). \end{aligned} \tag{25}$$

Define $\psi(s) = \sum_a \mu(a|s) \sum_{s', r} (r + \gamma V_m^{\mu}(s')) \cdot \nabla_{\theta} P_{\theta}(s', r|s, a)$, and additionally define $\rho_{\theta}^{\mu}(s \to x, n)$ as the transition probability under the policy $\mu$ from state $s$ to $x$ after $n$ steps in the learned MDP model $P_{\theta}$. Then, we can rewrite the above equation as:

$$\begin{aligned} \nabla_{\theta} V_m^{\mu}(s) &= \psi(s) + \gamma \sum_{s'} \rho_{\theta}^{\mu}(s \to s', 1) \nabla_{\theta} V_m^{\mu}(s') \\ &= \psi(s) + \gamma \sum_{s'} \rho_{\theta}^{\mu}(s \to s', 1) \Big[ \psi(s') + \gamma \sum_{s''} \rho_{\theta}^{\mu}(s' \to s'', 1) \nabla_{\theta} V_m^{\mu}(s'') \Big] \\ &= \psi(s) + \gamma \sum_{s'} \rho_{\theta}^{\mu}(s \to s', 1) \psi(s') + \gamma^2 \sum_{s''} \rho_{\theta}^{\mu}(s \to s'', 2) \nabla_{\theta} V_m^{\mu}(s'') \\ &= \sum_{s'} \sum_{t=0}^{\infty} \gamma^t \rho_{\theta}^{\mu}(s \to s', t) \psi(s'), \end{aligned} \tag{26}$$

where the last equation of Eq. (26) is obtained by continuing to unroll $\nabla_{\theta} V(\cdot)$.

Substituting Eq. (26) into Eq. (23), we have

$$
\begin{aligned}
\nabla_\theta \mathbb{E}_{s\sim\rho_0} & \left[ \left( V_d^\mu(s) - V_m^\mu(s) \right)^2 \right] \\
= & -2\mathbb{E}_{s\sim\rho_0} \left[ \left( V_d^\mu(s) - V_m^\mu(s) \right) \nabla_\theta V_m^\mu(s) \right] \\
= & -2\mathbb{E}_{s\sim\rho_0} \left[ \left( V_d^\mu(s) - V_m^\mu(s) \right) \sum_{s'} \sum_{t=0}^\infty \gamma^t \rho_\theta^\mu(s \to s', t) \psi(s') \right] \\
= & -2\sum_{s\in\mathcal{S}} \rho_0(s) \left( V_d^\mu(s) - V_m^\mu(s) \right) \sum_{s'} \sum_{t=0}^\infty \gamma^t \rho_\theta^\mu(s \to s', t) \psi(s') \\
= & -2\sum_{s\in\mathcal{S}} \sum_{s'\in\mathcal{S}} \rho_0(s) \left( V_d^\mu(s) - V_m^\mu(s) \right) d_\theta^\mu(s, s') \psi(s'),
\end{aligned}
\tag{27}
$$

where $d_\theta^\mu$ is the (improper) discounted state transition probability from $s$ to $s'$. Note that from the definition of $\psi(s)$, we have

$$
\begin{aligned}
\psi(s) &= \sum_a \mu(a|s) \sum_{s',r} (r + \gamma V_m^\mu(s')) \cdot \nabla_\theta P_\theta(s', r|s, a) \\
&= \sum_a \mu(a|s) \sum_{s',r} (r + \gamma V_m^\mu(s')) P_\theta(s', r|s, a) \frac{\nabla_\theta P_\theta(s', r|s, a)}{P_\theta(s', r|s, a)} \\
&= \mathbb{E}_{a\sim\mu, (r,s')\sim P_\theta} \left[ (r + \gamma V_m^\mu(s')) \nabla_\theta \log P_\theta(s', r|s, a) \right].
\end{aligned}
$$

Therefore, we have

$$
\psi(s') = \mathbb{E}_{a'\sim\mu, (r',s'')\sim P_\theta} \left[ (r' + \gamma V_m^\mu(s'')) \nabla_\theta \log P_\theta(s'', r'|s', a') \right]
$$

Plugging $\psi(s')$ into Eq. (27), we obtain

$$
\begin{aligned}
\nabla_\theta \mathbb{E}_{s\sim\rho_0} \left[ \left( V_d^\mu(s) - V_m^\mu(s) \right)^2 \right] = & -2\sum_{s\in\mathcal{S}} \sum_{s'\in\mathcal{S}} \rho_0(s) \left( V_d^\mu(s) - V_m^\mu(s) \right) d_\theta^\mu(s, s') \psi(s') \\
= & -2\mathbb{E}_{s\sim\rho_0, s'\sim d_\theta^\mu, a'\sim\mu, (r',s'')\sim P_\theta} \left[ \left( V_d^\mu(s) - V_m^\mu(s) \right) \cdot \right. \\
& \left. (r' + \gamma V_m^\mu(s'')) \nabla_\theta \log P_\theta(s'', r'|s', a') \right].
\end{aligned}
$$

Hence, we finish the proof. □

## B. Proof of Propositions

**Proof of Proposition 4.1**

*Proof.* We want to show there is exactly one function $V^*$ satisfying

$$
V^*(s) = \widehat{T}_d^\mu V^*(s)
$$

for any state $s$, which means $V^*$ is the unique fixed point of $\widehat{T}_d^\mu$.

In the following, we prove that $\widehat{T}_d^\mu$ is a $\gamma$-contraction with respect to the supremum norm. Let $\|V\|_\infty = \sup_s |V(s)|$ denote the sup norm of a value function $V$. We will show

$$
\|\widehat{T}_d^\mu V - \widehat{T}_d^\mu W\|_\infty \leq \gamma \|V - W\|_\infty \quad \text{for all } V, W.
$$

Fix any state $s$, we consider the difference $\widehat{T}_d^\mu V(s) - \widehat{T}_d^\mu W(s)$.

If $|\mathcal{D}_s| > 0$, then

$$\widehat{T}_d^\mu V(s) = \frac{1}{|\mathcal{D}_s|} \sum_{(s,a,r,s') \in \mathcal{D}_s} \left[ r + \gamma V(s') \right],$$

$$\widehat{T}_d^\mu W(s) = \frac{1}{|\mathcal{D}_s|} \sum_{(s,a,r,s') \in \mathcal{D}_s} \left[ r + \gamma W(s') \right].$$

It follows that

$$\widehat{T}_d^\mu V(s) - \widehat{T}_d^\mu W(s) = \frac{\gamma}{|\mathcal{D}_s|} \sum_{(s,a,r,s') \in \mathcal{D}_s} \left[ V(s') - W(s') \right].$$

Therefore, we have

$$\left| \widehat{T}_d^\mu V(s) - \widehat{T}_d^\mu W(s) \right| = \frac{\gamma}{|\mathcal{D}_s|} \left| \sum_{(s,a,r,s') \in \mathcal{D}_s} \left[ V(s') - W(s') \right] \right|$$

$$\leq \frac{\gamma}{|\mathcal{D}_s|} \sum_{(s,a,r,s') \in \mathcal{D}_s} \left| V(s') - W(s') \right|.$$

Since $\left| V(s') - W(s') \right| \leq \|V - W\|_\infty$ for all $s'$, it holds that

$$\left| \widehat{T}_d^\mu V(s) - \widehat{T}_d^\mu W(s) \right| \leq \frac{\gamma}{|\mathcal{D}_s|} \sum_{(s,a,r,s') \in \mathcal{D}_s} \|V - W\|_\infty = \gamma \|V - W\|_\infty.$$

If $|\mathcal{D}_s| = 0$, by definition, we have

$$\left| \widehat{T}_d^\mu V(s) - \widehat{T}_d^\mu W(s) \right| = 0 \leq \gamma \|V - W\|_\infty.$$

Overall, for any state $s$, we have

$$\left| \widehat{T}_d^\mu V(s) - \widehat{T}_d^\mu W(s) \right| \leq \gamma \|V - W\|_\infty.$$

Taking the supremum over $s$ yields

$$\|\widehat{T}_d^\mu V - \widehat{T}_d^\mu W\|_\infty = \sup_s \left| \widehat{T}_d^\mu V(s) - \widehat{T}_d^\mu W(s) \right| \leq \gamma \|V - W\|_\infty.$$

Hence $\widehat{T}_d^\mu$ is indeed a $\gamma$-contraction under the sup norm.

By the Banach Fixed Point Theorem, any $\gamma$-contraction ($0 \leq \gamma < 1$) on a complete normed vector space has a unique fixed point. Therefore, there exists a unique $V^*$ such that

$$V^*(s) = \widehat{T}_d^\mu V^*(s), \quad \forall s.$$

We denote $V^*(s)$ by $V_d^\mu(s)$ which completes our proof. □

### Proof of Proposition 4.2

*Proof.* This follows the same argument as the standard proof of the Bellman operator's fixed-point uniqueness, substituting the true model $P$ with the learned model $P_\theta$. □

## C. Limitation

While VIPO explicitly leverages the discrepancy between $V_d^\mu$ and $V_m^\mu$ to achieve substantial improvements in datasets with narrow coverage, its performance is subject to certain limitations. Under conditions of extremely poor dataset coverage where data is entirely insufficient to yield an informative $V_d^\mu$, the value inconsistency regularizer loses its reliable anchor, which may potentially bias the model. Furthermore, if the neural network lacks sufficient representational capacity to approximate the true dynamics accurately, value inconsistency cannot be fully minimized. This resulting irreducible error bounds the potential improvement that VIPO can achieve over standard negative log-likelihood (NLL) training.

---

**Algorithm 2** Planner

---

1: **Require:** Offline dataset $\mathcal{D} = \{(s_i, a_i, r_i, s'_i)\}_{i=1}^N$; approximate dynamics model $P_\theta$ learned from Algorithm 1; critics $\{Q_{\psi_1}, Q_{\psi_2}\}$
2: Initilize the replay buffer $\mathcal{D}_m \leftarrow \emptyset$
3: **while** not reaching maximum iterations $T$ or convergence criterion **do**
4:     Generate $h$-step rollouts by $P_\theta$ and add them to $\mathcal{D}_m$
5:     Sample a mini-batch $B = \{s, a, r, s'\}$ from $\mathcal{D} \cup \mathcal{D}_m$
6:     Compute the target values for $B$ according to Eq. (32) and Eq. (33)
7:     Learn the optimal control policy $\pi_\phi$ according to Eq. (34)
8: **end while**
9: Output the optimized policy parameter $\phi^* = \phi$

---

## D. Planner details

VIPO is a general framework designed to systematically enhance model accuracy and can be seamlessly integrated into existing model-based offline RL algorithms. To ensure a fair comparison, we retain the original planner of the base method when integrating VIPO. In this section, we detail the MOBILE planner in Algorithm 2, which is used to obtain the results for VIPO-MOBILE. For VIPO-MOPO and VIPO-LEQ, we utilize the standard planners described in Yu et al. (2020) and Park & Lee (2024), respectively.

To implement Algorithm 1, we adopt the objective function defined in Janner et al. (2019) as the original model loss. The specific form of the original model loss is expressed as:

$$\mathcal{L}_{ori}(\theta) = -\mathbb{E}_\mathcal{D}\left[\log P_\theta(s', r|s, a)\right] \tag{28}$$

For instance, we might define our model predictive model $P_\theta$ to produce a Gaussian distribution with diagonal covariances, parameterized by $\theta$ and conditioned on $s$ and $a$, i.e.: $P_\theta(s', r|s, a) = \mathcal{N}(\boldsymbol{\mu}_\theta(s, a), \boldsymbol{\Sigma}(s, a))$. Then the original model loss has the following form:

$$\mathcal{L}_{ori}^G(\theta) = \sum_{n=1}^N [\boldsymbol{\mu}_\theta(s_n, a_n) - s_{n+1}]\boldsymbol{\Sigma}_\theta^{-1}(s_n, a_n)[\boldsymbol{\mu}_\theta(s_n, a_n) - s_{n+1}] + \log\det\boldsymbol{\Sigma}_\theta(s_n, a_n). \tag{29}$$

Given a pre-trained environment model $P_\theta$ generated by Algorithm 1, the agent simulates $h$-step rollouts starting from the state in $\mathcal{D}$ in the learned model $P_\theta$ and then stores these synthetic transitions to the replay buffer $\mathcal{D}_m$. For policy training, we incorporate the uncertainty quantification $\mathcal{U}(s, a)$ with the soft actor-critic (SAC) algorithm (Haarnoja et al., 2018). Specifically, we use the uncertainty quantification based on Bellman Inconsistency proposed in Sun et al. (2023):

$$\begin{aligned}\mathcal{U}(s, a) &= \text{Std}\left(P_\theta^i Q_\psi(s, a)\right) \\ &= \text{Std}\left(\gamma\mathbb{E}_{\substack{\{s'_j\}\sim P_\theta^i \\ \{a'_j\}\sim\pi}}\left[\min_{k=1,2}\bar{Q}_{\psi_k}(s'_j, a'_j)\right]\right),\end{aligned} \tag{30}$$

where $\bar{Q}_{\psi_k}$ is the target state-action value network obtained by an exponentially moving average of parameters of the state-action value network $Q_{\psi_k}$. The objective function for critics is defined as:

$$\mathcal{L}_{critic} = \mathbb{E}_{(s,a,r,s')\sim\mathcal{D}\cup\mathcal{D}_m}\left[(Q_{\psi_k} - y)^2\right], \tag{31}$$

where the target value for $(s, a, r, s') \in \mathcal{D}$ is

$$y = r + \gamma\left[\min_{k=1,2}\bar{Q}_{\psi_k}(s', a') - \alpha\log\pi_\phi(a'|s')\right], \tag{32}$$

and the target for $(s, a, r, s') \in \mathcal{D}_m$ is

$$y = r + \gamma\left[\min_{k=1,2}\bar{Q}_{\psi_k}(s', a') - \alpha\log\pi_\phi(a'|s')\right] - \beta\mathcal{U}(s, a). \tag{33}$$

*Table 5.* Comparisons of system-level hyperparameters under different policy backends.

| Components | MOPO | MOBILE | LEQ |
|---|---|---|---|
| Training scheme | | MBPO-styled ensemble Gaussian dynamics with value-inconsistency loss | |
| Conservatism | Model uncertainty | Lower confidence bound | Lower expectile |
| Policy | Stochastic | Stochastic | Deterministic |
| Policy objective | $Q(s, a)$ | $Q(s, a)$ | $\lambda$-returns |
| $\lambda$ | 1e-6 or 1e-9 | 1e-6 or 1e-9 | 1e-6 or 1e-9 |
| # of dynamics model | 5 or 7 | 7 | 7 |
| # of critics | 2 | 1 | 1 |
| Critic objective | One-step | One-step | $\lambda$-returns |
| Horizon length ($H$) | 10 | 1 | 10 |
| Expansion length ($R$) | 1 | 5 | 5 |
| Discount rate ($\gamma$) | 0.99 | 0.99 | 0.997 |
| $\beta$ | 1.0 | 0.95 | 0.25 |

The policy is optimized by solving the following optimization problem:

$$\pi_\phi = \max_\phi \mathbb{E}_{\substack{\sim \mathcal{D} \cup \mathcal{D}_m \\ a \sim \pi_\phi}} \left[ \min_{k=1,2} Q_{\psi_k}(s, a) - \alpha \log \pi_\phi(a|s) \right] \tag{34}$$

# E. Experimental Details

### E.1. Experimental setups

**D4RL.** The D4RL Mujoco tasks serve as a benchmark for evaluating offline RL algorithms in continuous control environments, such as Hopper, Walker2d, and HalfCheetah. These tasks employ datasets of varying quality, including random, expert, and mixed trajectories, to evaluate agents' ability to derive effective policies from offline data. This setup provides a standardized framework for assessing offline RL performance in continuous control domains.

**NeoRL.** NeoRL's MuJoCo tasks provide offline RL benchmarks in environments such as HalfCheetah-v3, Walker2d-v3, and Hopper-v3. These tasks are based on conservative datasets generated from suboptimal policies, mimicking real-world scenarios characterized by limited and narrowly distributed data. This setup poses a challenge for algorithms to learn effective policies from constrained offline datasets, promoting the development of methods applicable to practical settings. Notably, NeoRL offers varying numbers of trajectories for training data across tasks: 100, 1,000, and 10,000. For our experiments, we uniformly selected 1,000 trajectories for each task.

**AntMaze.** The D4RL AntMaze tasks benchmark offline RL in sparse-reward navigation with long-horizon decision making. The agent must navigate a maze to reach a goal from offline datasets collected by different behavior policies, including *play* and *diverse* variants that differ in trajectory coverage and start–goal distributions. We consider multiple maze layouts with increasing difficulty, ranging from medium and large mazes to the *ultra* setting. In particular, AntMaze-Ultra features more complex maze geometries and uneven terrain, leading to harder state transitions and larger distributional shift between the offline data and the optimal policy. Overall, these characteristics make AntMaze, especially the ultra variant, a challenging benchmark for evaluating robustness and generalization in offline navigation.

**Baselines.** In D4RL Gym tasks, we evaluate VIPO against various offline RL algorithms, including model-free methods such as TD3+BC (Fujimoto & Gu, 2021), which trains a policy subject to a behavior cloning constraint. We also compare VIPO to CQL (Kumar et al., 2020), which calculates conservative Q-values for out-of-distribution (OOD) samples, and IQL (Kostrikov et al., 2021), which learns optimal policies offline through expectile regression on the dataset to address extrapolation errors. Additionally, we consider DQL (Wang et al., 2022), which utilizes diffusion models to effectively generate optimal actions from offline datasets. In the model-based category, we include MOPO (Yu et al., 2020), which incorporates uncertainty-aware dynamics models to penalize OOD state-action pairs; MOReL (Kidambi et al., 2020), which constructs uncertainty-aware, penalized MDPs to restrict policy learning to in-distribution states; RAMBO (Rigter et al., 2022), which uses adversarial models to handle distributional shifts in offline data; and MOBILE (Sun et al., 2023), which applies Bellman-inconsistency uncertainty quantification for robust offline policy learning.

For the NeoRL tasks, we compare VIPO against CQL, TD3+BC, EDAC, MOPO, and MOBILE, where EDAC (An et al.,

*Table 6.* Task-specific hyperparameters.

| Domain | Task | Hyperparam ($\eta$ / $\tau$) |
|---|---|---|
| | halfcheetah-r | $\eta$= 0.05 |
| | hopper-r | $\eta$= 0.05 |
| | walker2d-r | $\eta$= 0.5 |
| | halfcheetah-m | $\eta$= 0.05 |
| | hopper-m | $\eta$= 0.05 |
| | walker2d-m | $\eta$= 0.5 |
| D4RL Gym | halfcheetah-mr | $\eta$= 0.05 |
| | hopper-mr | $\eta$= 0.05 |
| | walker2d-mr | $\eta$= 0.5 |
| | halfcheetah-me | $\eta$= 0.05 |
| | hopper-me | $\eta$= 0.05 |
| | walker2d-me | $\eta$= 0.5 |
| | HalfCheetah-L | $\eta$= 0.05 |
| | Hopper-L | $\eta$= 0.5 |
| | Walker2d-L | $\eta$= 0.5 |
| | HalfCheetah-M | $\eta$= 0.05 |
| NeoRL | Hopper-M | $\eta$= 0.5 |
| | Walker2d-M | $\eta$= 0.5 |
| | HalfCheetah-H | $\eta$= 0.05 |
| | Hopper-H | $\eta$= 0.5 |
| | Walker2d-H | $\eta$= 0.5 |
| | umaze | $\tau$= 0.1 |
| | umaze-diverse | $\tau$= 0.1 |
| AntMaze | large-play | $\tau$= 0.2 |
| | large-diverse | $\tau$= 0.1 |
| | ultra-play | $\tau$= 0.1 |
| | ultra-diverse | $\tau$= 0.1 |

2021) addresses value overestimation by decoupling the target policy from the behavior policy.

For AntMaze tasks, we compare VIPO against MOBILE, IQL-TD-MPC (Chitnis et al., 2024), and LEQ (Park & Lee, 2024). IQL-TD-MPC extends TD-MPC to the offline setting, utilizing a hierarchical latent planning mechanism to generate high-level intents that guide the policy while LEQ induces conservatism via lower expectile regression of $\lambda$-returns to enable low-bias model-based value estimation.

### E.2. Benchmarks

We perform experiments on Gym tasks (v2 version) included in the D4RL (Fu et al., 2020) benchmark. In addition, we leverage the NeoRL benchmark, which offers a more challenging evaluation setting that closely resembles real-world scenarios, to provide a more comprehensive assessment of offline RL algorithms. NeoRL tasks are constructed using conservative datasets generated from suboptimal policies, reflecting real-world conditions characterized by limited and narrowly distributed data. This design poses substantial challenges for algorithms to learn effective policies from constrained offline datasets, thereby driving the development of approaches better suited for practical applications.

The following sections outline the sources of the reported performance on these benchmarks.

**D4RL.** (1) For MOPO* (Yu et al., 2020), as the original paper reported results on "v0" datasets, we reference the experimental results provided in (Sun et al., 2023), which are based on the "v2" datasets. For CQL (Kumar et al., 2020), we report the scores obtained on the "v2" datasets using the codebase available at `github.com/yihaosun1124/OfflineRL-Kit`. (2) For TD3+BC (Fujimoto & Gu, 2021), MOReL (Kidambi et al., 2020), COMBO (Yu et al., 2021), RAMBO (Rigter et al., 2022), and MOBILE (Sun et al., 2023), we directly cite the performance results reported in their respective original papers, as these studies evaluated Gym tasks using the "v2" datasets (refer to Table 1). (3) For IQL (Kostrikov et al., 2021) and DQL (Wang et al., 2022), we conducted experiments on the "v2" random datasets using the codebase provided by the authors of the respective papers. For other "v2" datasets, the results are taken directly from their original publications.

*Table 7.* Normalized average returns on D4RL Adroit tasks, averaged over 4 random seeds.

| Task Name | BC | CQL | TD3+BC | MOPO | MOBILE | VIPO-MOBILE |
|---|---|---|---|---|---|---|
| *pen-human* | $25.8 \pm 8.8$ | $35.2 \pm 6.6$ | -1.0 | 10.7 | $30.1 \pm 14.6$ | $\mathbf{52.6 \pm 7.7}$ |
| *door-human* | $2.8 \pm 0.7$ | $1.2 \pm 1.8$ | -0.2 | -0.2 | $-0.2 \pm 0.1$ | $2.0 \pm 0.3$ |
| *hammer-human* | $\mathbf{3.1 \pm 3.2}$ | $0.6 \pm 0.5$ | 0.2 | 0.3 | $0.4 \pm 0.2$ | $1.1 \pm 0.9$ |
| *pen-cloned* | $38.3 \pm 11.9$ | $27.2 \pm 11.3$ | -2.1 | 54.6 | $69.0 \pm 9.3$ | $\mathbf{71.1 \pm 9.5}$ |
| *hammer-cloned* | $0.7 \pm 0.3$ | $1.4 \pm 2.1$ | -0.1 | 0.5 | $1.5 \pm 0.4$ | $\mathbf{2.1 \pm 0.2}$ |
| **Average** | $14.1 \pm 5.0$ | $13.1 \pm 4.5$ | -0.6 | 13.2 | $20.2 \pm 4.9$ | $25.8 \pm \mathbf{3.7}$ |

**NeoRL.** The performance results for CQL, and MOPO are sourced from the original NeoRL paper. For TD3+BC and EDAC, we report the scores retrained by the authors of Sun et al. (2023).

### E.3. Hyperparameters

Table 5 and Table 6 summarize the hyperparameter configurations used in our experiments. We report both system-level settings that define the training and rollout schemes of different policy backends, and task-specific parameters that adapt the method to dataset characteristics. Unless explicitly stated, all hyperparameters are shared across tasks.

Across all backends, we adopt an MBPO-style ensemble Gaussian dynamics model trained with value-inconsistency regularization. The primary differences between MOPO, MOBILE, and LEQ arise from how conservatism is enforced and how policies are optimized. MOPO and MOBILE employ stochastic policies with one-step $Q(s, a)$ objectives, whereas LEQ uses a deterministic policy trained with $\lambda$-returns. These choices induce distinct rollout structures, with LEQ favoring longer horizons and expansion lengths, and MOBILE relying on short-horizon rollouts. Other system-level configurations, including ensemble size, number of critics, discount factor $\gamma$, and penalty coefficient $\beta$, follow the backend-specific settings in Table 5.

For task-dependent hyperparameters, the real-data ratio $\eta$ is adjusted for D4RL Gym and NeoRL tasks to balance real and model-generated data. A consistent trend is observed across domains: HalfCheetah tasks prefer smaller $\eta$, while Hopper and Walker2d benefit from higher real-data ratios, particularly under lower-quality datasets. For AntMaze tasks, we instead adjust the temperature parameter $\tau$ to control conservatism in sparse-reward, long-horizon navigation, with larger and more complex mazes requiring stronger regularization, as summarized in Table 6.

All experiments are conducted on NVIDIA RTX 4090 GPUs.

## F. Experiments on more complex manipulation tasks

The Adroit benchmark presents a set of high-dimensional manipulation tasks using a 24-DoF simulated robotic hand. The tasks include writing with a pen, hammering a nail, and opening a door, each requiring fine-grained control and coordination. We consider two types of datasets: "human", which consists of 25 expert demonstrations collected from real human teleoperation, and "cloned", a 50-50 mixture of these demonstrations and data generated by a cloned policy trained on them.

In Table 7, we report the evaluation results of VIPO on the Adroit benchmark from D4RL. The inherent complexity of the Adroit benchmark arises from its high-dimensional observation and action spaces combined with relatively sparse demonstration data. VIPO addresses these challenges by incorporating a value-informed loss into the dynamics model training, encouraging consistency between the predicted values under the learned dynamics and the empirical returns from the dataset. This promotes more value-aligned dynamics without explicitly optimizing the policy.

The results illustrate that VIPO consistently outperforms several established methods, including model-free algorithms (BC, CQL, TD3+BC) and prominent model-based approaches (MOPO, MOBILE) across multiple benchmark tasks (see Table 7). Specifically, notable performance gains are observed in tasks such as pen-human (52.6 ± 7.7), pen-cloned (71.1 ± 9.5), and hammer-cloned (2.1 ± 0.2), underscoring VIPO's capacity to manage the intricacies associated with high-dimensional manipulation scenarios.

