# OpenReview forum: "VIPO: Value Function Inconsistency Penalized Offline Reinforcement Learning"
_ICML.cc/2026/Conference — ICML 2026 regular_

### Official Review · Reviewer_Qc2o · 2026-02-21

**Soundness:** 3
**Presentation:** 2
**Significance:** 3
**Originality:** 3
**Overall Recommendation:** 4
**Confidence:** 4

**Summary:**

This paper proposes VIPO, a new framework for model-based offline reinforcement learning that aims to reduce the reliance on heuristic and potentially unreliable uncertainty-driven conservatism. The key idea is to obtain two estimates of the behavior-policy value function via dual usage of offline data.

**Compliance With Llm Reviewing Policy:**

Affirmed.

**Final Justification:**

Thanks for the authors' rebuttal. I have no further questions.

**Key Questions For Authors:**

The description of algorithms such as VIPO- in the experiment section is unclear. For example, the authors mention that VIPO-MOBILE is replacing the models used in previous methods with the model trained by VIPO. The method in this paper and MOBILE's model training method are different, so why can the model be directly replaced? Do the authors mean using the ensemble from MOBILE, but the training method for the critic and actor is VIPO? I believe the description at this point is unclear.

There are no references to the appendix in the main text, which is not conducive to understanding the main body of the article, especially since some derivation processes are not friendly to most readers, such as Theorem 3.5 and Theorem 3.4.

I hope the authors provide training curves for halfcheetah-m, halfcheetah-mr, walker2d-mr, hopper-mr, and hopper-m, as well as the corresponding checkpoints and evaluation code.

**Limitations:**

No. The authors have not provided sufficient discussion of limitations.

**Strengths And Weaknesses:**

Strengths: This paper has a sufficient theoretical foundation, comprehensive experiments, and good model performance

Weaknesses: The description of the method section needs to be further clarified, and there are vague descriptions in the experimental section

---

> ### Author Rebuttal · Authors · 2026-03-31
>
> We thank the reviewer for the positive assessment and for the helpful comments on clarity and reproducibility. Below, we address your specific questions and concerns point by point.
>
> **Q1: Clarification of VIPO-MOBILE / VIPO-MOPO / VIPO-LEQ**
>
> We apologize that this part was stated too briefly. VIPO is a *model-learning framework*, rather than a new planner. In VIPO-MOBILE, we keep the downstream MOBILE pipeline unchanged, including rollout generation, uncertainty handling, critic update, and actor update, and only replace the *dynamics model learner* with our VIPO objective. Concretely, the learned model still provides the same dynamics-model interface $P_{\theta}(s', r \mid s, a)$ consumed by MOBILE, so it can be directly plugged into the original MOBILE backend without modifying the actor/critic training procedure. The same principle applies to VIPO-MOPO and VIPO-LEQ: we replace only the model-learning module and retain the base planner. We agree that Section 4.1 currently makes this point insufficiently explicit, and we will revise Section 4.1 and Appendix D accordingly.
>
> **Q2: Missing references to the appendix**
>
> We agree. In the revision, we will add explicit pointers from the main text to Appendix A (related work), Appendix B/C (proofs of Theorems 3.4/3.5 and propositions), Appendix D (planner details), and Appendix E (experimental setups and hyperparameters), so that readers can more easily follow both the derivations and the implementation details.
>
> **Q3: Training curves, checkpoints, and evaluation code**
>
> We appreciate this helpful suggestion. Under the rebuttal-stage link policy and double-blind constraints, external links may only be used for anonymous figures/tables and their captions, and reviewers are not required to follow them. Therefore, at this stage, we are only able to provide the requested training/reward curves for *halfcheetah-m*, *halfcheetah-mr*, *walker2d-mr*, *hopper-mr*, and *hopper-m* through anonymous external figures/tables. As for the corresponding checkpoints and evaluation code, we will further organize them and release them once the review process has concluded and the anonymity constraints are lifted. To view these requested training curves, please refer to our anonymous external repository at: **https://anonymous.4open.science/r/ICML-RE-VIPO-582F/README.md**
>
> **Limitations**
>
> We also agree that the limitation discussion should be made explicit. We will add a dedicated paragraph noting that VIPO is a plug-in framework for improving dynamics learning rather than replacing planner-side components, and that it introduces additional training overhead because $V_d$ and $V_m$ are learned together with the dynamics model.

---

> > ### Author Rebuttal · Reviewer_Qc2o · 2026-04-02
> >
> > Thanks for your rebuttal. My concerns have been well addressed.

---

> > > ### Author Response · Authors · 2026-04-02
> > >
> > > We sincerely thank the reviewer for acknowledging that all concerns have been fully addressed. We truly appreciate your positive assessment and the continued endorsement of our work. Any further support in your final evaluation would be deeply appreciated.

---

### Official Review · Reviewer_GtKR · 2026-03-09

**Soundness:** 3
**Presentation:** 3
**Significance:** 3
**Originality:** 3
**Overall Recommendation:** 5
**Confidence:** 3

**Summary:**

This paper proposes VIPO, an offline model-based reinforcement learning method that trains the dynamics model with an additional objective encouraging consistency between a value estimated from offline data and a value induced by the learned model. The method is designed to be modular and compatible with existing offline MBRL pipelines rather than replacing their planning components. Experiments on D4RL, NeoRL, and AntMaze suggest that this value-guided model training strategy can improve performance and model quality in many settings.

**Compliance With Llm Reviewing Policy:**

Affirmed.

**Final Justification:**

The rebuttal addressed my main concerns.

**Key Questions For Authors:**

1 Can you clarify how much bias is introduced by replacing the model-induced visitation distribution with the empirical data distribution in the practical objective?

2 How sensitive is the method to the weighting of the value inconsistency term, and how stable is this sensitivity across different benchmark families?

3 To what extent do the reported gains come from learning a genuinely better dynamics model rather than from being particularly compatible with the specific planners used in the experiments?

4 Can you compare against more recent strong offline or model-based RL baselines, or explain why the current comparison set is sufficient to establish competitiveness?

In settings with poor data coverage, how reliable is the data-driven value target used for calibration, and have you observed cases where it can misguide model learning?

**Limitations:**

yes

**Strengths And Weaknesses:**

Strengths

1 The paper introduces a clear and intuitive idea of using value inconsistency to directly regularize dynamics learning in offline MBRL.

2 The method appears complementary to prior approaches such as uncertainty-aware planning methods because it targets model training rather than the downstream planner.

3 The empirical study covers several benchmark suites and includes analyses beyond final return, such as prediction error and uncertainty-related behavior.

4 The proposed framework is presented as a practical plug-in module that can be integrated into existing model-based offline RL pipelines.

Weaknesses

1 There is still a noticeable gap between the theory and the practical algorithm because the implementation replaces the model-induced visitation distribution with an off-policy approximation without fully analyzing the resulting bias or its consequences.

2 The ablation study is not yet strong enough to cleanly isolate where the gains come from, including the contribution of the inconsistency loss itself, its sensitivity to weighting, and its interaction with the downstream planner.

3 The empirical comparison would be more convincing with stronger recent baselines from the latest offline and model-based RL literature.

4 The improvements are promising but not uniformly consistent across all tasks, especially on more difficult settings where the method appears to require careful regularization.

---

> ### Author Rebuttal · Authors · 2026-03-31
>
> We sincerely thank the reviewer for the positive assessment and the informative suggestions. Below, we address your specific questions and concerns point by point.
>
> **W1 \& Q1: Clarification on the Off-Policy Surrogate Gradient**
>
> We agree that Eq. (12) uses a surrogate gradient rather than the exact gradient in Theorem 3.5. More precisely, what is approximated is the sampling distribution: the exact gradient requires expectations under the model-induced visitation distribution, whereas our algorithm replaces this with the empirical data distribution to achieve a fully offline, numerically stable update. This replacement introduces an *off-policy gradient bias* governed by the distribution mismatch. Let $\nu_\theta^\mu$ denote the exact tuple distribution in Theorem 3.5, $\hat{\nu}\_D$ denote the empirical tuple distribution in Eq. (12), and $g_\theta(z)$ denote the per-sample gradient integrand. Then:
>
> $$\nabla\_\theta L\_{\mathrm{vic}}^{\mathrm{exact}}(\theta) = \mathbb{E}\_{z\sim \nu\_\theta^\mu}[g\_\theta(z)], \qquad \nabla\_\theta L\_{\mathrm{vic}}^{\mathrm{surr}}(\theta) = \mathbb{E}\_{z\sim \hat{\nu}\_D}[g\_\theta(z)].$$
>
> The induced bias is exactly:
>
> $$\mathrm{Bias}(\theta) = \nabla\_\theta L\_{\mathrm{vic}}^{\mathrm{surr}}(\theta) - \nabla\_\theta L\_{\mathrm{vic}}^{\mathrm{exact}}(\theta) = \mathbb{E}\_{z\sim \hat{\nu}\_D}[g\_\theta(z)] - \mathbb{E}\_{z\sim \nu\_\theta^\mu}[g\_\theta(z)].$$
>
> If $\|g_\theta(z)\|\le G$, applying the triangle inequality yields:
> $$\|\mathrm{Bias}(\theta)\| \le 2G\mathrm{TV}(\hat{\nu}\_D,\nu\_\theta^\mu) \le 2G\Big( \mathrm{TV}(\hat{\nu}\_D,\nu^\mu) + \mathrm{TV}(\nu^\mu,\nu\_\theta^\mu) \Big),$$
>
> where $\nu^\mu$ denotes the true behavior-policy tuple distribution. This decomposition makes the approximation error explicit: the first term is the empirical sampling error, and the second is the mismatch between the model-induced and true behavior-policy visitation distributions. Thus, the bias remains small when the dataset approximates the behavior occupancy well and the learned model is reasonably accurate. We emphasize that this approximation is deliberate: computing exact model-induced visitations at every step is expensive and amplifies compounding errors early in training. Grounding the update on empirical data yields a stable surrogate while preserving the VIC signal.
>
> **W2 \& Q2 \& Q3: VIC Ablation, Sensitivity, and Gain Attribution**
>
> Our experiments isolate the source of these gains in two ways. First, at the model level, Table 4 directly compares the original-loss model (OL Model) against the VIPO-trained model, showing consistently lower validation prediction errors for VIPO. This proves the VIC objective directly improves model fidelity. Second, at the policy level, VIPO replaces *only the model-learning module* while keeping the downstream planners (MOPO, MOBILE, LEQ) entirely unchanged. The fact that VIPO systematically improves performance across three entirely different planner backends confirms that the gains stem from a genuinely better dynamics model, rather than specific planner compatibility.
>
> Regarding the weighting coefficient $\lambda$, it is not a brittle, task-specific tuning knob. Instead, it serves to balance the scale of the original likelihood term and the VIC term. In practice, we calibrate $\lambda$ at the *domain/benchmark-family level*. Once the order of magnitude matches the reward scale of a given family, the method is highly stable; as shown in Table 5, we use only a very small set of $\lambda$ choices across all benchmarks.
>
> **W3 \& Q4: Baseline Choice**
>
> We thank the reviewer for highlighting these recent concurrent works, which we will certainly discuss in the revision (e.g., MOMO, ADMPO-OFF, NEUBAY). To address this immediately, we have evaluated our method against these recent baselines and provided an updated comparison table at the following anonymous link: **https://anonymous.4open.science/r/ICML-RE-VIPO-582F/README.md**
>
> **W4 \& Q5: Hard Settings and Poor Data Coverage**
>
> We agree that extreme poor data coverage is a natural limitation. If the behavior data is near-random or coverage is exceptionally sparse, the dataset is insufficient to yield a reliable $V_d^\mu$. In such cases, the value inconsistency regularizer loses its informative anchor, and enforcing alignment may propagate value estimation errors into the model dynamics. However, VIPO is designed for practical offline RL regimes where sufficient data exists to extract a meaningful behavior-policy value signal. In these regimes, Theorem 3.4 guarantees asymptotic consistency, and empirically, the method remains robust without requiring heavy per-task tuning. For instance, using our family-level hyperparameters, VIPO improves over MOBILE on all 18 medium/hard tasks across D4RL and NeoRL. We will formally add this discussion on data coverage boundaries to the Limitations section.

---

> > ### Author Rebuttal · Reviewer_GtKR · 2026-04-01
> >
> > Thank you for the helpful and detailed rebuttal, it has largely addressed my main concerns, and I will raise my score accordingly.

---

> > > ### Author Response · Authors · 2026-04-01
> > >
> > > Thank you very much for your positive feedback and for acknowledging our efforts in the rebuttal. We are glad to hear that your main concerns have been adequately addressed.
> > >
> > > We will faithfully incorporate our discussions and the suggested improvements into the final revision of the manuscript to further enhance its quality.
> > >
> > > We truly appreciate your support and your decision to raise the score, and we look forward to the updated evaluation. Thank you again for your time and expertise.

---

### Official Review · Reviewer_w69x · 2026-03-13

**Soundness:** 4
**Presentation:** 3
**Significance:** 3
**Originality:** 3
**Overall Recommendation:** 4
**Confidence:** 3

**Summary:**

This paper introduces VIPO, an offline model-based learning algorithm which learns the value function and next step value function using a surrogate gradient objective. The authors theoretically justify this and provide experiments which show god performance of VIPO, using certain model based policy algorithms.

**Compliance With Llm Reviewing Policy:**

Affirmed.

**Final Justification:**

The authors sufficiently addressed concerns, and reinforced my positive assessment of the paper.

**Key Questions For Authors:**

- How does this algorithm compare with respect to runtime?
- $\bar V$ is an EMA of the parameters, but this is not much discussed.
- Tab. 4 shows better model prediction, but does not show pessimism, which $\bar M$ is supposed to be. Can this be further discussed and measured?

**Limitations:**

no limitations discussed.

**Strengths And Weaknesses:**

**Strengths**
- This work, and its design decisions are well justified by historical RL theory and well justified.
- The overall algorithm and loss are well justified, and explained.
- Further, this work demonstrates strong performance on most tasks, compared to other algorithms.
- The main ablation, showing increased uncertainty with increased drop rate, is also nice.

**Weaknesses**
- As with model based algorithms, this requires an extra step of policy extraction, which can slow runtime. Runtime comparison or discussing would be desirable.
- More measurements regarding how the dynamics model generalizes outside of the support of the data and pessimism would be desirable.
- VIPO is not well contextualized in terms of work on improving dynamics models.

---

> ### Author Rebuttal · Authors · 2026-03-31
>
> We sincerely thank the reviewer for the positive assessment and for recognizing both the theoretical motivation and the empirical strength of our method. Below, we address your specific questions and concerns point by point.
>
> **W1 \& Q1 Runtime and policy extraction**
>
> We thank the reviewer for this question. We would like to clarify that VIPO is a *model-learning framework*, rather than a new planner. In VIPO-MOPO / VIPO-MOBILE / VIPO-LEQ, we retain the original policy-extraction component of the corresponding base method, and only replace the dynamics model learner. Therefore, relative to the base model-based offline RL backend, VIPO does not introduce additional planner-side runtime or an extra policy-extraction phase. The additional cost is concentrated in the *model-training stage*, since $V_d$ and $V_m$ are newly introduced ingredients of our method. In practice, this overhead is modest in our implementation; with early stopping, the dynamics-model training typically takes about 30--40 minutes on a single RTX 4090 GPU. We will clarify this distinction in the revision so that the runtime trade-off is presented more precisely.
>
> **Q2: EMA target networks**
>
> We thank the reviewer for pointing this out. Exponential moving average (EMA) target networks are a standard and widely used stabilization strategy for bootstrapped Bellman regression, and we also considered them in this spirit in our implementation. At the same time, we intentionally avoided introducing EMA when it was not necessary. In practice, we found that for sparse-reward tasks such as AntMaze, directly updating the value networks was already sufficiently stable, so EMA was not needed there. By contrast, for dense-reward MuJoCo tasks, EMA remained helpful for stabilizing the value targets during training. Overall, even when EMA is used, the additional overhead is negligible, since the value networks are lightweight. We will clarify this implementation detail more explicitly in the revision.
>
> **W2 \& Q3: Generalization outside data support and pessimism**
>
> We thank the reviewer for this important point. We agree that Table 4 by itself is not intended to be a direct measure of pessimism. Rather, this concern should be viewed jointly through Figure 2 and Table 4, which serve complementary purposes in our paper. Specifically, Figure 2 (Section 4.2) evaluates the uncertainty-aware aspect of our method: when the available data are progressively reduced, the uncertainty estimated by VIPO increases consistently with the drop ratio, whereas MOPO remains largely insensitive. This indicates that VIPO better captures the uncertainty induced by data scarcity. In contrast, Table 4 (Section 4.3) evaluates predictive capability and shows that VIPO learns a more accurate dynamics model with lower validation prediction error. Therefore, Figure 2 provides evidence from the uncertainty/pessimism perspective, while Table 4 provides evidence from the model-accuracy perspective. Taken together, these two experiments can address the reviewer’s concern from both sides. We will revise the text to make this logical connection more explicit.
>
> **W3: on improving dynamics models**
>
> We thank the reviewer for this helpful suggestion. We respectfully note that the current draft already discusses representative model-based offline RL methods such as MOPO / MOReL / MOBILE in the main text and RAMBO / LEQ in the appendix. However, we agree that the specific thread of *improving the dynamics model itself* could be organized more explicitly in the main text, so that the positioning of VIPO is clearer. In the revision, we will make this structure more explicit by distinguishing methods that mainly impose conservatism at the planner/downstream stage after standard model fitting from methods that modify the model-learning stage itself, and then clarifying that VIPO belongs to the latter category. In particular, unlike planner-side pessimistic adjustment, VIPO improves the learned dynamics model during training by aligning the model-induced behavior-policy value with a data-grounded behavior-policy value learned from the offline dataset, i.e., through long-horizon value-consistency alignment. We will add this discussion more explicitly in the main text and provide clearer pointers to the appendix.
>
> **Limitation**
>
> We thank the reviewer for pointing out this omission. We agree that our limitations should be stated explicitly. Following the reviewer’s suggestion, we will add that VIPO introduces modest additional training overhead because $V_d$ and $V_m$ are learned together with the dynamics model, although it does not add extra planner-side complexity beyond the base model-based backend.

---

> > ### Author Rebuttal · Reviewer_w69x · 2026-04-03
> >
> > I thank the authors for their rebuttal. It has sufficiently addressed my concerns and I maintain my positive score.

---

> > > ### Author Response · Authors · 2026-04-04
> > >
> > > Thank you for your time and for acknowledging our rebuttal. We deeply appreciate your continued support and your positive evaluation of our work.

---

### Official Review · Reviewer_gU8H · 2026-03-13

**Soundness:** 2
**Presentation:** 3
**Significance:** 2
**Originality:** 3
**Overall Recommendation:** 5
**Confidence:** 4

**Summary:**

VIPO introduces a model-based offline reinforcement learning framework that improves model accuracy by leveraging a self-supervised value function inconsistency loss. VIPO exploits the dataset in two complementary ways. First, it directly estimates the behavior policy value from data and separately from the learned model. Second, it uses this discrepancy as a regularization signal. This dual usage guides the model toward more accurate dynamics and better reflects epistemic uncertainty, overcoming the unreliability of ensemble-based uncertainty estimates in previous methods. Empirically, VIPO consistently outperforms prior offline RL approaches and achieves state-of-the-art performance across D4RL and NeoRL benchmarks. Overall, VIPO provides a general, easily integrated framework for improving model-based offline RL.

**Compliance With Llm Reviewing Policy:**

Affirmed.

**Final Justification:**

I thank the authors for their detailed rebuttal. My concerns have been addressed and have revised my score accordingly.

**Key Questions For Authors:**

The paper describes two key differences from prior work: (1) the dual usage of the dataset to learn both approximated value function
of the behavior policy and the dynamics model, and (2) the value function inconsistency loss used during model training. However, algorithmically it seems that the dual usage only affects model learning through the VIC loss. Could the authors clarify whether these are intended as separate contributions? In particular, is there a meaningful variant that uses the dual-usage setup but does not include the VIC regularization?

**Limitations:**

The paper does not explicitly discuss the limitations of VIPO in scenarios with poor dataset coverage or misspecified models. Including such a discussion would strengthen the contribution.

**Strengths And Weaknesses:**

**Strengths:**

- Using value function inconsistency as a training signal for model learning is intuitive, well-motivated, and conceptually novel in the offline RL context.

- The framework is modular and broadly compatible with existing model-based offline RL methods.

- The algorithmic design is coherent and relatively easy to integrate into existing model-based RL pipelines, with clear procedures for computing VIC and combining it with NLL losses.

- The empirical evaluation is extensive, covering D4RL, NeoRL, and AntMaze benchmarks.

**Weaknesses:**

- Related work is relegated to the appendix. While space may be tight, properly positioning the contribution is important because VIPO is primarily a framework rather than a fundamentally new algorithm.

- The paper describes two conceptual contributions (dual usage of the dataset and value inconsistency regularization) but algorithmically these do not appear to be separable. The dual usage arises only as a mechanism to compute the value inconsistency loss. Consequently, it is unclear whether these should be considered distinct contributions

- VIPO's value inconsistency regularization assumes an accurate dataset-derived value function, but the paper does not analyze how errors in this estimate might affect the model training.

- The theoretical results establish fixed-point properties and a generalization bound for the augmented objective but do not formally demonstrate that minimizing the value inconsistency term improves model accuracy or offline policy performance compared to likelihood alone.

- Some improvements over baselines appear modest and may fall within the reported standard deviations for several tasks.

---

> ### Author Rebuttal · Authors · 2026-03-31
>
> We sincerely thank the reviewer for the time and effort dedicated to reviewing our manuscript. Your constructive feedback and insightful comments have been highly valuable in helping us improve the quality of our paper. Below, we provide a detailed, point-by-point response to address each of your concerns.
>
> **W1: Positioning of Related Work**
>
> We agree that appropriately positioning the related work is essential. Upon acceptance, we will leverage the extra page allowance to move this section to the main text, immediately following the Introduction.
>
> **W2 \& Q1: Dual Usage and VIC Regularization**
>
> We do not intend to claim “dual-usage” as a standalone contribution or that it independently yields performance gains. Instead, it serves as a *generic* descriptive term for our approach, reflecting a more effective use of data. This is analogous to referring to the VIC loss as “self-supervised” learning, which does not imply that self-supervision alone accounts for the observed improvements.
>
> Our contribution is the general VIPO framework *as a whole* to improve model learning accuracy.
>
> Regarding the variant of our approach, it is possible to develop different "dual-usage" methods together with different loss functions, to potentially improve the model learning accuracy. However, it is too broad under the term "dual-usage" (like "self-supervision" method) to guide further investigation.
>
> **W3: Value-Estimate Error**
>
> We would like to clarify that (1) VIPO *does not* assume the dataset-derived value function $V_d^\mu$ is accurate, and (2) the impact of value estimation error on model learning error is explicitly analyzed in Theorem 3.4, which provides a formal bound on the induced error. This finite-sample estimation error enters the bound through the "alignment bias'' term, $\bigl(\sqrt{\tfrac{\lambda}{2},\mathcal{L}_{\mathrm{vic}}(\theta^\star)}\bigr)$.
>
> Specifically, at the true model $\theta^\star$, the alignment bias is exactly the error between the learned value compared to the ground truth value:
>
> $$\mathcal{L}\_{\mathrm{vic}}(\theta^\star) = \mathbb{E}\_{s \sim \rho_0} [ (V\_d^\mu(s) - V^\mu(s))^2].$$
>
> This quantifies how $V_d^\mu$'s accuracy affects the learned model's accuracy, which is a constant bias.
>
> **W4: Clarification on Theoretical Results**
>
> Theorem 3.4 is indeed a direct model-accuracy result: it explicitly upper bounds the Total Variation distance between the learned model $P_{\theta_{\mathrm{opt}}}$ and the true dynamics $P^\star$, revealing how optimizing $L_{\mathrm{aug}}(\theta)$ controls model error.
>
> We acknowledge that our theorem is not a pairwise dominance result stating VIPO must outperform likelihood-only estimators in all finite-sample settings, nor does it bound downstream policy performance. Rather, it establishes error control for VIPO's augmented loss.
>
> This comparative advantage is established empirically. In Section 4.1, replacing only the dynamics model while keeping the planner fixed yields return gains attributable to improved model quality. In Section 4.3 (Table 4), VIPO reduces predictive error from 0.173 to 0.121 (a 30\% reduction) compared to the original-loss baseline.
>
> **W5: Magnitude of Improvements**
>
> While the absolute numerical gains on certain individual tasks might appear modest, we would like to highlight that they represent statistically significant improvements that exceed the reported standard deviations.
>
> Regarding standard deviations, for several tasks, our improvements fall well outside the standard deviation margins, demonstrating statistical significance. VIPO-MOBILE achieves gains of +5.4 on halfcheetah-m (std 0.4), +5.7 on hopper-m-r (std 0.9), and +8.5 on walker2d-m-r (std 0.3). In NeoRL, the average score improves drastically (60.7 to 73.3).
>
> Furthermore, consistent cross-benchmark gains indicate algorithmic superiority over random noise. VIPO-MOBILE improves over MOBILE on all 12 D4RL Gym tasks (average 80.0 to 83.6), and VIPO-LEQ improves over LEQ (61.0 to 64.9) on AntMaze. Because these gains are obtained by replacing only the model-learning module, keeping the planner fixed, they provide robust evidence of the regularizer's effectiveness.
>
> **Limitation: Poor Dataset Coverage and Misspecification**
>
> We appreciate this constructive suggestion and will formally include this discussion. VIPO explicitly leverages the $V_d^\mu$ vs. $V_m^\mu$ discrepancy, which is why it demonstrates substantial improvements in datasets with narrow coverage. However, under extremely poor dataset coverage, if data is entirely insufficient to yield an informative $V_d^\mu$, the value inconsistency regularizer loses its reliable anchor, potentially biasing the model.
>
> Furthermore, if the neural network lacks sufficient representational capacity to approximate the true dynamics accurately, value inconsistency cannot be fully minimized. This irreducible error bounds the potential improvement that VIPO can achieve over standard NLL training.

---

> > ### Author Rebuttal · Reviewer_gU8H · 2026-04-03
> >
> > I thank the authors for their detailed rebuttal. My concerns have been addressed and have revised my score accordingly.

---

> > > ### Author Response · Authors · 2026-04-04
> > >
> > > Thank you very much for your time, your constructive feedback, and for acknowledging our rebuttal. We deeply appreciate your support for our work and your revised score.

---

### Decision · Program_Chairs · 2026-04-30

**Decision:**

Accept (regular)

**Comment:**

This study's fundamental theme pertains to advancing offline model-based reinforcement learning. The authors demonstrate that existing approaches prioritize training models primarily through negative log-likelihood (NLL) loss, often failing to account for how these models perform when applied to downstream value learning. To address this, the authors introduce an additional loss function that refines the dynamics model by minimizing the discrepancy between value functions estimated via the model and those learned directly from the offline dataset. The authors provide theoretical analysis and empirical results to demonstrate that this framework is highly generalizable. The consensus among the reviewers is that this work provides new, valuable insights into the field of offline model-based reinforcement learning. The manuscript's important theme concerns the theoretical limitations identified during the review process. Specifically, I feel that propositions 3.1 and 3.2 rely on a strong assumption: that the Bellman mapping remains within the value function class. There is a concern that if the value function lacks sufficient expressivity, the resulting errors could be exacerbated. Furthermore, I feel that the paper lacks a detailed discussion of certain previous works, such as the approach presented in [1] which also attempts to correct learned models through value alignment. Consequently, refining the manuscript to address these theoretical assumptions and explicitly contextualize the research within existing literature would significantly strengthen the work.

[1] Mastering Atari, Go, Chess and Shogi by Planning with a Learned Model